# An Information Theoretic Approach to Machine Unlearning

**Jack Foster**                                    *jwf40@cam.ac.uk*
*Department of Engineering*
*University of Cambridge, UK*

**Kyle Fogarty**                                   *ktf25@cam.ac.uk*
*Department of Computer Science and Technology*
*University of Cambridge, UK*

**Stefan Schoepf**                                 *ss2823@cam.ac.uk*
*Department of Engineering*
*University of Cambridge, UK*

**Zack Dugue**                                     *zdugue@caltech.edu*
*Department of Computer Science*
*California Institute of Technology, United States*

**Cengiz Öztireli**                                *aco41@cam.ac.uk*
*Department of Computer Science and Technology*
*University of Cambridge, UK*

**Alexandra Brintrup**                             *ab702@cam.ac.uk*
*Department of Engineering*
*University of Cambridge, UK*

**Reviewed on OpenReview:** *https://openreview.net/forum?id=t1utIThKHD*

## Abstract

To comply with AI and data regulations, the need to forget private or copyrighted information from trained machine learning models is increasingly important. The key challenge in unlearning is forgetting the necessary data in a timely manner, while preserving model performance. In this work, we address the zero-shot unlearning scenario, whereby an unlearning algorithm must be able to remove data given only a trained model and the data to be forgotten. We explore unlearning from an information theoretic perspective, connecting the influence of a sample to the information gain a model receives by observing it. From this, we derive a simple but principled zero-shot unlearning method based on the geometry of the model. Our approach takes the form of minimising the gradient of a learned function with respect to a small neighbourhood around a target forget point. This induces a smoothing effect, causing forgetting by moving the boundary of the classifier. We explore the intuition behind why this approach can jointly unlearn forget samples while preserving general model performance through a series of low-dimensional experiments. We perform extensive empirical evaluation of our method over a range of contemporary benchmarks, verifying that our method is competitive with state-of-the-art performance under the strict constraints of zero-shot unlearning.

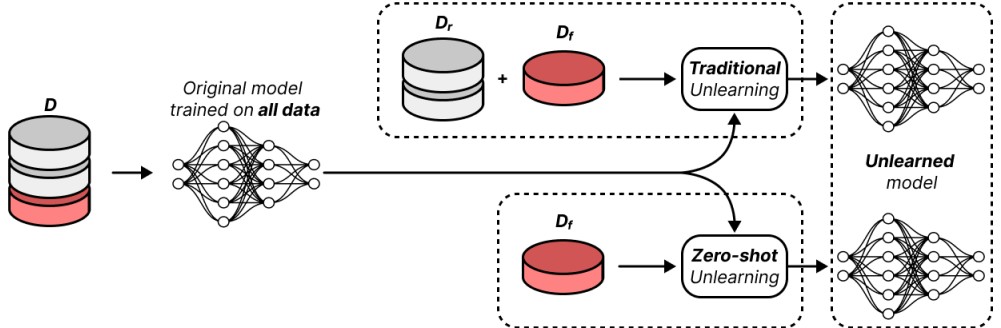

Figure 1: Visualization of the zero-shot unlearning scenario. Contrary to traditional unlearning there is no access to, or prior knowledge of, any data other than the forget set or the model at any point beyond its current state. These constraints make the problem considerably more challenging.

# 1 Introduction

Regulations such as the General Data Protection Regulation (GDPR) enshrine an individual's data autonomy rights, including the right to be forgotten. While deleting an entry from a database is relatively straightforward, removing the influence of that data from a trained model is a challenging open problem. The process of removal, referred to as *unlearning*, is difficult for several reasons. It is known that neural networks memorise instance-level information (Arpit et al., 2017; Zhang et al., 2021; Feldman, 2020), and it is practically intractable to ascribe parameter changes to a specific training sample post-hoc (Kurmanji et al., 2023). At its core, unlearning is a multi-objective optimization problem with three key desiderata. An effective unlearning algorithm must remove the influence of the selected subset of data, maintain model performance on retained data, and minimise computational cost. These goals are antagonistic since inducing forgetting inevitably disrupts the model's performance, balancing these objectives is key. Naïvely, one may achieve perfect forgetting by retraining a model on the training data sans the forget samples every time there is a forget request. However, this is prohibitively expensive thus violating the third desideratum.

Many of the unlearning methods proposed are effective, however they make strong assumptions about the problem setting that simplifies the task considerably. Primarily, existing methods typically assume access to all, or a subset of, the training data. This data is used in different ways, such as to fine-tune the model post-forgetting (Chundawat et al., 2023a; Graves et al., 2021), or to conduct parameter importance calculations after the initial training period (Foster et al., 2023). In reality, there are many reasons why this data could be unavailable, such as cost of storage, limited duration access to datasets, or an oversight in considering machine unlearning during model development. As such, Chundawat et al. (2023b) introduce a novel problem setting for unlearning, termed *zero-shot* (ZS) unlearning, whereby no data is available to the unlearning model. A limitation of this problem definition is that, without a concrete way to define the forget set, we are constrained to forgetting full class only. As such, we explore the zero-shot setting that permits an arbitrary forget set to be defined, but importantly only the data to be forgotten and the trained model are available (Figure 1). This is extremely challenging, since the remaining data is not available to protect model performance, and thus more delicate methods are required. Insightful treatment of the ZS scenario can be found in Chen et al. (2023), where unlearning is formulated as reconstructing a decision boundary that could be reasonably learnt by a model trained without the forget data, achieved through learning the nearest false label for each forget sample.

In this work, we approach ZS unlearning from an information theoretic perspective. Golatkar et al. (2020) consider information leakage when observing model weights, whereas we consider the information gained by a model by training on a given sample. Data points offer a classifier different amounts of information gain when included in the training data (Lindley, 1956; Houlsby et al., 2011). If a data point can be inferred from other training data, then it offers little information gain (Jeong and Qiu, 2018). In ZS unlearning, knowing the contribution a sample has made to a model is hard as we have access to only the model and the sample to be forgotten. We postulate that the information gain of a sample can serve as a good indicator for how much

influence it has over a model. From this hypothesis, we derive a principled loss to induce forgetting based on how the geometry of the problem space changes with respect to the softmax classifier. If a sample offers little information gain then it may be inferred from other data, therefore the classifier's output should change minimally over similar samples. This minimal change can be effectively measured via the rate of change of the model with respect to the input, which should be low in the region surrounding low-information samples. In contrast, a high information gain sample cannot be inferred, and therefore one would expect they lie in high rate of change regions of space. We present Just in Time (JiT) unlearning, a novel ZS unlearning algorithm based on minimising the gradients of a classifier with respect to local neighbourhood around each forget sample. We show in low dimensional experiments that removing training samples from low-gradient regions yields minimal changes to the learned decision boundary, whereas removing samples from high gradient regions has a more pronounced effect. We therefore propose that, with reference to Feldman (2020), samples with low information gain may be predicted with more generalised knowledge and thus do not infringe on privacy. In contrast, samples with large information gain have significant impact on the learned classifier, are more likely memorised, and do infringe on privacy. We demonstrate empirically that following these principles, JiT causes the removal of influence from the forget set while preserving generalisation performance across the wider space.

Our primary contributions are as follows:

- To the best of our knowledge, JiT is the first unlearning algorithm to be directly informed by the information gain of a sample.
- We provide extensive empirical analysis of the geometry of JiT unlearning in low dimensions.
- We show our method is competitive with existing SOTA in the zero-shot domain.

## 2 Related Work

**Information theory** is concerned with the transmission, quantification, and storage of information (Shannon, 1948), and has seen widespread use in machine learning. Most relevant here is its use in determining the information gain of an experiment (Lindley, 1956). This notion has seen uses such as determining splits in decision trees (Quinlan, 1986), and active learning (Tong and Koller, 2001; Houlsby et al., 2011). Here we use this concept as a proxy to a training sample's influence on a learned function.

**Machine unlearning** was first introduced in Cao and Yang (2015), and a probabilistic perspective of unlearning was explored in Ginart et al. (2019); Sekhari et al. (2021); Gupta et al. (2021); Neel et al. (2021); Triantafillou et al. (2023). Here we focus on post-hoc unlearning methods that operate on models that are already trained. Methods exist that alter the initial training scheme (Bourtoule et al., 2021; Mehta et al., 2022; Shah et al., 2023), but these are considered out of scope as they do not satisfy the ZS problem constraints.

Current SOTA methods rely on accessing all or a subset of the original dataset that is *not* to be forgotten (i.e. the retain set), thus violating the ZS constraints. Bad Teacher unlearning (Chundawat et al., 2023a) and SCRUB (Kurmanji et al., 2023) leverage a student-teacher framework while Amnesiac unlearning (Graves et al., 2021) trains with randomised labels for forget data before fine-tuning on the retained data to repair the model. UNSIR (Tarun et al., 2023) learns an error-maximising noise to induce forgetting of the necessary data, before also employing a finetuning step. Warnecke et al. (2021) minimise the divergence in model output over a sample and its noisy perturbations and then finetune. A key limitation of these methods is that protecting model performance necessitates access to retain-set data for the entire duration of the model's lifetime. To address this, Golatkar et al. (2020) and Foster et al. (2023) propose methods that do not require fine-tuning or repair steps. Golatkar et al. (2020) derives an unlearning algorithm that minimises information gained about the training data when observing model weights. However, this scales quadratically with dataset size and often performs considerably worse than state-of-the-art (Tarun et al., 2023; Foster et al., 2023). Selective Synaptic Dampening is a scalable retrain-free approach, based on inducing forgetting by selectively dampening parameters that are disproportionately important to the forget-set (Foster et al., 2023). This requires access to the whole dataset at least once, to calculate the importance over the retained data. Chundawat et al. (2023b) introduces two methods to address ZS unlearning. The first method, an extension of Tarun et al. (2023), replaces the repair step with an error-minimising noise. The second utilises a generator network and an attention loss to distil knowledge from an expert teacher, with a band-pass filter preventing

the flow of knowledge for specific classes. Both methods are slow, do not scale well to large problem spaces, and can only forget entire classes. Chen et al. (2023) present boundary shrinking and boundary expanding. Shrinking causes unlearning by training over the nearest false label for forget samples, found via a fast gradient sign attack (Goodfellow et al., 2014). While performant, shrinking scales poorly with model and input size. Boundary expanding is faster but less performant, remapping forget samples by training them to fit a new output neuron, before removing the neuron leaving the forget samples in high entropy states.

We note that many recent works have focused on generative models such as LLMs. Most evaluations in this space consider unlearning in the context of question answering (QA) rather than classification, for example, the TOFU benchmark focuses on QA about fictitious authors (Maini et al., 2024). The WMDP benchmark features four multiple choice questions about weapons of mass destruction (Li et al., 2024). The same work introduces Representation Misdirection Unlearning, which trains the LLM to produce random unit vectors as intermediate embeddings when prompted with hazardous queries, while simultaneously finetuning over a retain set to preserve wider knowledge. Negative preference optimization is an LLM unlearning mechanism that can be viewed as a stable formulation of gradient ascent over the forget data (Zhang et al., 2024). An important nuance of forgetting in LLMs is that forgetting needs to apply to a sequence of tokens, but the constituent tokens need to remain accessible to the model, thus making the domains fairly distinct, however Li et al. (2024) does show that SCRUB and SSD can generalize to LLMs reasonably. Despite LLMs receiving recent attention, classification remains a cornerstone of deep learning; its prevalance in both large-scale and sensitive problems in industry ranging from social-media to healthcare necessitates the continued research of unlearning in this domain.

**Membership inference attacks (MIA)** are a way of measuring information leakage of a machine learning model (Shokri et al., 2017). An auxiliary model (e.g. a logistic regression) is trained to infer whether a given data point was included in a model's training data. MIAs are used as typical evaluation metrics in machine unlearning; if a MIA cannot recognise a forgotten sample as an element from the train set, then this presents empirical evidence that the sample has indeed been forgotten.

**Randomized smoothing** constructs an adversarially robust classifier by adding random noise to the input during training (Cohen et al., 2019), making the model less sensitive to small perturbations in the input space. This has clear connections to the unlearning method we present in this work, which induces forgetting through a post-hoc, localized variation of randomized smoothing.

## 3 Preliminaries

We introduce the notation for machine unlearning in a supervised classification task, consistent with the approach outlined in Chen et al. (2023). Consider some input space $\mathcal{X} \subset \mathbb{R}^d$ and some output label space $\mathcal{Y} \subset \mathbb{R}^c$, where $d$ is the dimensionality of the input and $c$ is the number of classes. We define a training dataset $\mathcal{D} = \{\boldsymbol{x}_i, \boldsymbol{y}_i\}_{i=1}^N \subseteq \mathcal{X} \times \mathcal{Y}$, where $\boldsymbol{x}_i$ is a training input sample with the label $\boldsymbol{y}_i$. We denote a subset $\mathcal{D}_f \subseteq \mathcal{D}$ as the forget set, and $\mathcal{D}_r = \mathcal{D} \setminus \mathcal{D}_f$ as the retain set.

Let $f_\theta : \mathcal{X} \to \mathcal{Y}$ be a neural network, with parameters $\theta$. We assume that $f_\theta$ is well trained and generalises to in distribution test samples well. The objective of ZS unlearning is, given only the model $f_\theta$ trained on $\mathcal{D}$, to remove the influence of $\mathcal{D}_f$ from the learned model such that the unlearnt model $f_{\theta'}$ is approximately equivalent to a model retrained on only $\mathcal{D}_r$ which we define as the optimal solution, $f_{\theta^*}$. Since direct access to $f_{\theta^*}$ is, by definition of the unlearning problem, impossible, existing works construct approximate heuristics to induce forgetting and use a membership inference attack to measure forgetting. These attacks typically evaluate the difference in the output distributions of the model over train and test samples.

## 4 Proposed Method

In this section, we introduce our JiT unlearning method and provide intuition for its effectiveness by examining the geometry of a learned classifier and analyzing how the location of a forget sample in the input space can impact the decision boundary of a model retrained without it. This section considers the case where $|\mathcal{D}_f| = 1$, noting that larger subsets have an additive effect.

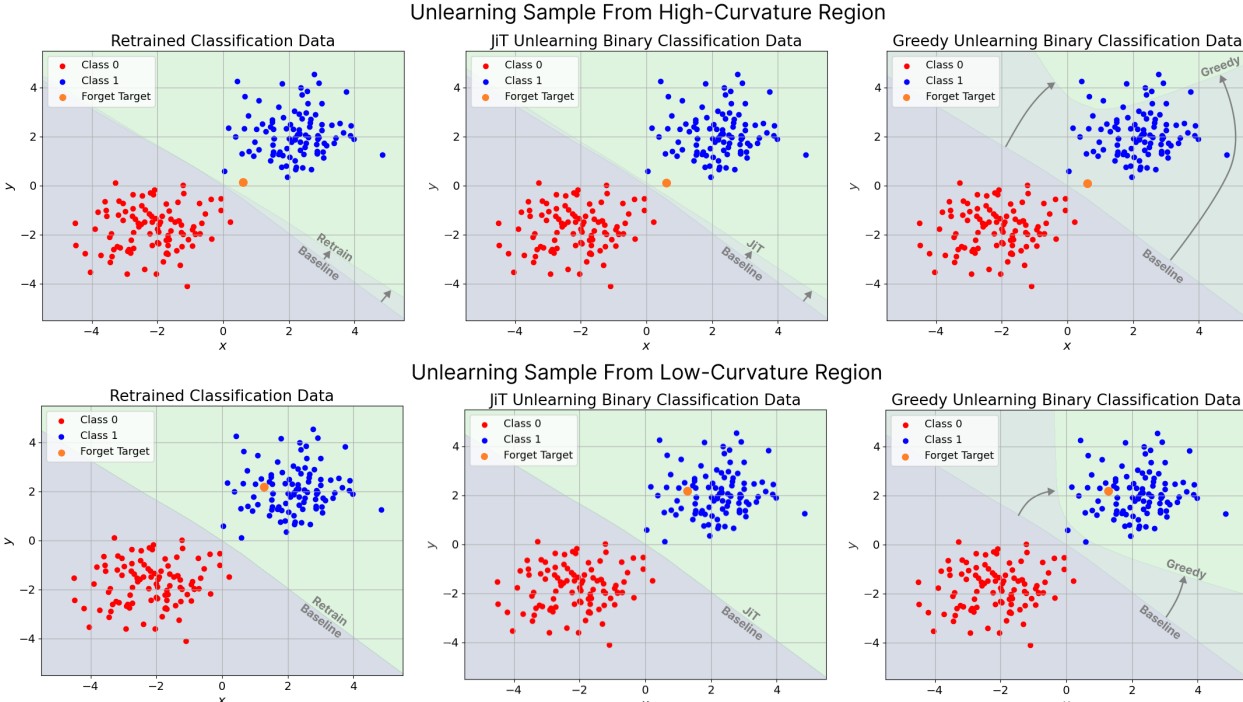

Figure 2: Demonstration of how the boundary of a classifier moves during unlearning. Retrained model is the gold standard. Removing a sample from a low-gradient region has almost no effect on the retrained model, whereas removing a sample from high gradient space has significant impact. In this low-dimensional setting, JiT successfully reconstructs the retrained boundary, whereas naively training to mislabel the forget sample completely destroys the trained model.

Consider the hypothetical of removing a single image of a black cat from a dataset comprised of 1 million black cats and 1 million white dogs. This likely has minimal effects on the learning process, since a well trained classifier should generalise and infer the class of the sample easily. If a sample may be removed from the training dataset without significant changes in the resultant model, we posit that an unlearning algorithm should also have minimal effect on the model when unlearning such a sample. As such, it is logical to design an unlearning algorithm that accounts for the information gain of a sample. However, directly measuring this quantity is difficult, especially in a ZS setting where there is no access to other data points. We therefore seek to derive a heuristic that can approximate how much information gain a sample may have offered the model, based on only that sample and the model itself. We begin by formally introducing the notion of a neighbourhood of a target sample:

**Definition 4.1** (Neighbourhood of a sample). Let $\mathcal{X} \subseteq \mathbb{R}^d$ be an input space of samples, and assume each sample $\mathbf{x} \in \mathcal{X}$ is associated with a class label $c(\mathbf{x}) \in \{1, 2, ..., K\}$. Fix a radius r > 0. For each $\boldsymbol{x} \in \mathcal{X}$, define its neighbourhood $\mathcal{B}_r(\mathbf{x})$ as:

$$\mathcal{B}_r(\mathbf{x}) = \{\mathbf{z} \in \mathcal{X} : ||\mathbf{z} - \mathbf{x}|| \leq r\}.$$

Then,

**Definition 4.2** (Information of a sample). For each $\mathbf{x} \in \mathcal{X}$, define:

$$\alpha(\mathbf{x}) = \frac{1}{|\mathcal{B}_r(\mathbf{x})|} \sum_{\mathbf{z} \in \mathcal{B}_r(\mathbf{x})} \mathbb{I}\{c(\mathbf{z}) = c(\mathbf{x})\},$$

where $\mathbb{I}\{\cdot\}$ is the indicator function. In other words, $\alpha(\mathbf{x})$ is the proportion of $\mathbf{x}$'s neighbours that share its class label. Choose a threshold $\tau \in [0, 1]$. We say that a sample $\mathbf{x} \in \mathcal{X}$ is low information if $\alpha(\mathbf{x}) \geq \tau$, and high information otherwise.

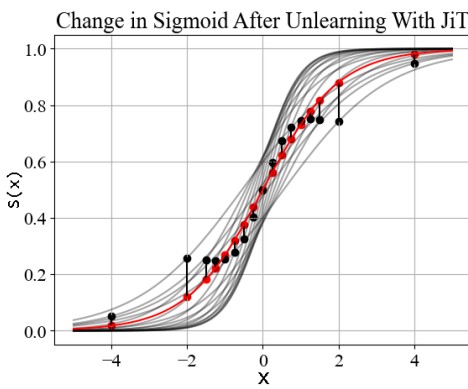
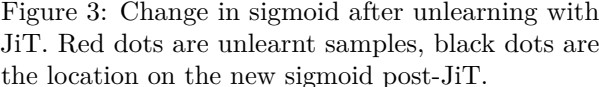
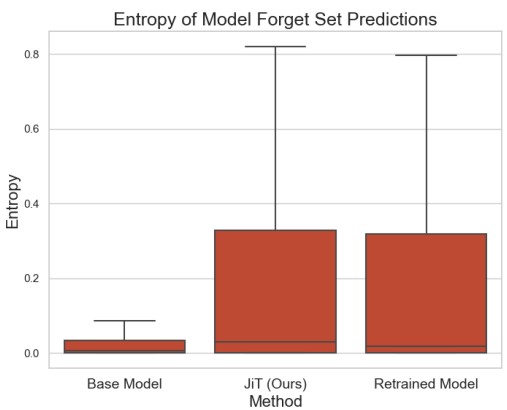

Figure 3: Change in sigmoid after unlearning with JiT. Red dots are unlearnt samples, black dots are the location on the new sigmoid post-JiT.

Figure 4: Entropy, $\mathcal{H}(x)$, of the $\mathcal{D}_f$ output distributions for full-class unlearning on CIFAR-10, showing JiT exhibits performance similar to the retrained model.

Plainly, we can say that a data point may be said to be low information if it can be inferred from its neighbourhood; and high information if it can not. Consider a low information training sample $x_l$, from definitions 4.1 and 4.2 we can expect that for some bound $r$, $f_\theta(\boldsymbol{x_l}) \approx f_\theta(\hat{\boldsymbol{x_l}}) \,\forall \hat{\boldsymbol{x_l}} \in \mathcal{B}_r(\boldsymbol{x_l})$. In other words, a model's predictions over a low information sample and a set of similar data should have similar output distributions. As such, the rate of change of the model in this space will be low. However, for a high information gain sample, this would not necessarily hold. From this we can describe an unlearning objective; if the classifier exhibits small output variation with respect to input perturbations around a forget sample (i.e., has small local gradients), then this sample likely has minimal influence on the decision boundary. Hence, we present a method based on minimising the gradient of the classifier with respect to input generated by perturbing points in the forget set. Since taking the gradient of the model with respect to the input is extremely expensive for larger problems, we instead construct a first order approximation to the gradient at the target via considering noisy perturbations within its neighbourhood. Formally, $\forall \boldsymbol{x} \in \mathcal{D}_f$, we seek to minimise the loss given below:

$$\ell = \mathbb{E}\left( \frac{\|f_\theta(\boldsymbol{x}) - f_\theta(\boldsymbol{x} + \boldsymbol{\xi})\|_2}{\|\boldsymbol{x} - (\boldsymbol{x} + \boldsymbol{\xi})\|_2} \right) \approx \frac{1}{N} \sum_{j=1}^{N} \left( \frac{\|f_\theta(\boldsymbol{x}) - f_\theta(\boldsymbol{x} + \boldsymbol{\xi}_j)\|_2}{\|\boldsymbol{\xi}_j\|_2} \right). \tag{1}$$

Where $\boldsymbol{\xi}$ is a noise vector of equivalent dimensionality to $\boldsymbol{x}$, and each component $\xi_i$ of $\boldsymbol{\xi}$ is independently drawn from a Gaussian distribution such that $\xi_i \sim \mathcal{N}(0, \sigma^2)$. For samples that are highly influential, minimising this loss will smooth the local region and remove its influence from the model. For low-information samples that are generalised knowledge, the neighbourhood will be rather smooth resulting in minimal changes. A full algorithm for JiT is given in the appendix 10.1.

### 4.1 A Geometric Interpretation of JiT

We now present JiT from a geometric perspective, providing insight into why it causes forgetting and how it protects the wider function. Consider a simple 2D classification task, as visualised in figure 2. We pose a simple question: does it make sense to forget all regions of space equally? Naturally, the answer is no. Completely forgetting a sample from within a low-gradient region of space would necessitate the forgetting of almost the entire class, even if they are not part of the forget set. Furthermore, removing a sample from this region would not have significant ramifications on a model retrained from scratch, nor is it likely infringing on the privacy of an individual. Unlearning in this instance often requires minimal alterations to the model. In contrast, a sample that lies in a high-gradient region may not only have significant influence over the position of the learned boundary, but may have been misclassified had it not been included in the training data. Figure 2 shows that this intuition holds in low dimensions; when the forget sample is within the

low-gradient region, a model retrained on $\mathcal{D}_r$ exhibits almost no change, whereas when the sample in $\mathcal{D}_f$ is in a high-gradient region, the boundary is shifted considerably. Enshrining such behaviour into a ZS unlearning algorithm is tantamount; as protection through fine-tuning or regularization is not possible, a ZS algorithm must be surgical in its forgetting methodology. Figure 2 shows that unlearning using JiT yields a classifier almost identical to the retrained model in this low-dimensional setting whereas greedily training over $\mathcal{D}_f$ with a false label causes complete destruction of the model.

The heuristic behind JiT's performance is based on the gradient field of the classifier. The crux of this rests upon the inherent non-linearity of neural networks. Typically well trained models exhibit sharp decision boundaries with a large rate of change, with flatter behaviour within the class (Fridovich-Keil et al., 2022). As such, given two unit noise vectors $\boldsymbol{\xi_i}, \boldsymbol{\xi_j}$, where $\boldsymbol{\xi_i}$ points towards the decision boundary and $\boldsymbol{\xi_j}$ points away, the gradient of the classifier between $\boldsymbol{x}$ and $\boldsymbol{x} + \boldsymbol{\xi_i}$ will be larger than for $\boldsymbol{x} + \boldsymbol{\xi_j}$. As such, minimising equation 1 for samples near a boundary will be biased towards moving the boundary *towards* $\boldsymbol{x}$. This has the consequence of increasing the uncertainty of the prediction and potentially changing the samples' predicted class. To further highlight this phenomena, figure 3 shows how unlearning forget samples (red dots) from a learned sigmoid function (red line) changes the learned function. Two things should be observed here: first, samples that lie in low gradient regions have relatively small changes and secondly, the updates to the function have the effect of pulling the the forget sample towards the centre of the sigmoid, which is the decision boundary. Unlearning in this way increases model uncertainty over forgotten samples, without destroying the wider function.

## 4.2 Entropy Similarity

In the previous section, we demonstrated how in low dimensions JiT can induce forgetting of a single sample in a way similar to retraining the model from scratch. Now, we demonstrate that the same loss can be used to forget arbitrary subsets $\mathcal{D}$ in higher dimensions, including full classes. Since visualising decision boundaries in high dimensions is challenging, we instead evaluate the entropy of the model output. We train a 2-layer CNN on the CIFAR-10 dataset, focusing specifically on the task of forgetting class 0. We compare the entropy of the unlearned model over class 0 with that of the original model, and a model retrained from scratch without class 0. Intuitively, low-entropy predictions indicate higher model confidence, and therefore we expect that JiT unlearning will increase the model's entropy, aligning it closely with that of the retrained model.

Figure 4 shows the entropy of the forget-set output distributions for a CNN trained on CIFAR-10. Our unlearning approach increases the entropy over the forget set, reducing the divergence between it and that of a model retrained from scratch on $\mathcal{D}_r$. In fact, under a Wilcoxon signed-rank test (Woolson, 2007), we find there is no statistically significant difference between the model unlearned with JiT and the retrained model for $p = 0.10$. JiT and retraining both increase the entropy over the forget set, suggesting the resultant models behave in a similar way, possessing less knowledge of the forget samples compared to the baseline model. Alongside matching the entropy over the forget set, JiT preserve model performance, as the unlearned model drops only 2% accuracy (From 99% to 97%) on $\mathcal{D}_r$. Our algorithm demonstrates promising characteristics that are indicative of an effective unlearning algorithm.

## 5 Experimental Setup

### 5.1 Benchmarks

We implement the same benchmarks from Foster et al. (2023), which are similar to that of Chundawat et al. (2023a), Golatkar et al. (2020) and Kurmanji et al. (2023). We run experiments 10 times, reporting the mean and standard deviation of these performances. Where classes or sub-classes are forgotten, we show performance over the same class/sub-class as in Foster et al. (2023); performance on additional classes can be found in the appendix (10.3). We perform a hyper-parameter search across a single forget class/sub-class, then use these parameters for all classes. This is more realistic, as it cannot be known *a priori* what future forget sets may be presented to the method. The reported $\mathcal{D}_r$ accuracy refers to accuracy over a test set of samples from the classes in $\mathcal{D}_r$.

**Unlearning scenarios:** Typically the three unlearning scenarios are: i) Full-class forgetting, where a full

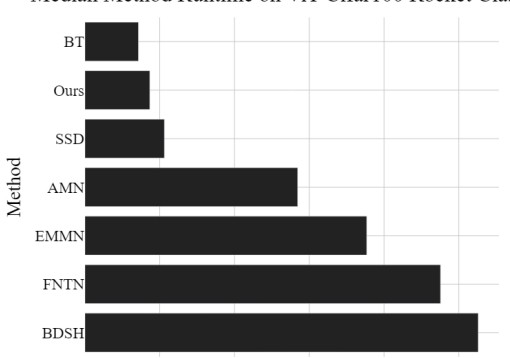

Figure 5: Median method runtime for ViT full-class forgetting on class rocket in seconds. For visual clarity we exclude GKT ($\sim$ 3000 seconds).

Table 1: VGG Full-class unlearning performance on PinsFaceRecognition class 1

| METHOD | $\mathcal{D}_r$ ACC. | $\mathcal{D}_f$ ACC. | MIA | ZS |
|---|---|---|---|---|
| BSLN | 94.0±0.0 | 93.9±0.0 | 13.82±0.0 | × |
| RTRN | 100.0±0.0 | 0.0±0.0 | 2.6±0.8 | × |
| FNTN | 97.6±0.7 | 36.9±9.9 | 4.3±2.7 | × |
| AMN | 99.7±0.1 | 0.0±0.0 | 1.4±1.33 | × |
| SCRUB | 98.8±0.0 | 97.1±0.0 | 8.8±0.76 | × |
| SSD | 55.8±0.0 | 0.0±0.0 | 4.0±0.0 | × |
| BT | 93.7±0.3 | 0.0±0.0 | 0.0±0.0 | × |
| UNSIR | 99.5±0.1 | 74.4±9.2 | 13.6±8.9 | × |
| GKT | 2.0±0.6 | 0.0±0.0 | 23.9±30.3 | ✓ |
| EMMN | 51.0±13.5 | 69.3±25.7 | 26.9±17.8 | ✓ |
| BDSH | 93.6±0.4 | 79.4±0.0 | 42.4±0.4 | ✓ |
| OURS | 91.4±0.1 | 1.9±0.2 | 4.7±0.5 | ✓ |

class from the dataset must be unlearned, ii) Sub-class forgetting, where a related subset from a class (e.g. all rockets from class vehicle) is forgotten, and iii) Random forgetting, where a subset is sampled uniformly from the entire training distribution. We evaluate our method in all three scenarios.

**Comparison methods:** We compare JiT against the following methods: i) *Baseline* (BSLN): that has not been unlearnt, ii) *Retrain* (RTRN): trained on only the retain data, iii) *Finetune* (FNTN): , where the model is fine-tuned on $\mathcal{D}_r$ for 5 epochs, iv) *Selective Synaptic Dampening* SSD (Foster et al., 2023), v) GKT (Chundawat et al., 2023b), vi) EMMN (Chundawat et al., 2023b), vii) SCRUB (Kurmanji et al., 2023), viii) *Bad Teacher* (BT) (Chundawat et al., 2023a) ix) *Amnesiac* (AMN) (Graves et al., 2021), x) UNSIR (Tarun et al., 2023), xi) *Boundary Shrinking* (BDSH) (Chen et al., 2023). Since GKT, EMMN, and UNSIR are theoretically limited to forgetting just a full-class, these cannot be evaluated in sub-class or random scenarios. Finally, we note that due to VRAM constraints, we could not benchmark SCRUB on ViT.

**Datasets:** As with previous work, we benchmark JiT on a range of image classification benchmarks. We make use of the CIFAR suite (Krizhevsky and Hinton, 2010), and the Pins Facial Recognition dataset (Burak, 2020), which consists of $17,534$ images of 105 celebrity faces.

**Models:** We evaluate methods on Vision Transformer (ViT) (Dosovitskiy et al., 2021) and VGG11 (Simonyan and Zisserman, 2014), trained on an NVidia RTX 4090 using Stochastic Gradient Descent with an initial learning rate of 0.1, and the OneCycle learning rate scheduler (Smith and Topin, 2019). Additionally, we compare the performance of JiT to BDSH on a ViT-L ($\sim 300m$ parameters) trained on the ILSVRC Imagenet dataset to demonstrate our method can scale to larger problem spaces.

**Evaluation metrics:** We evaluate model performance according to four key metrics: i) $\mathcal{D}_r$ accuracy, ii) $\mathcal{D}_f$ accuracy, iii) MIA score, and iv) method runtime. For all metrics bar runtime, the objective is not to minimise/maximise them, but rather to be as close to the retrained model as possible. This is important, as performing worse than the retrained model implies insufficient performance, but as noted in Chundawat et al. (2023a), Foster et al. (2023) and Kurmanji et al. (2023), significant deviation from the retrained model (e.g. *over-forgetting*) may leak information about the fact a sample has been forgotten. To remain consistent with existing unlearning literature we use the same logistic regression MIA evaluation as Chundawat et al. (2023a) and Foster et al. (2023).

**JiT hyper-parameters:** We conduct a hyper-parameter search for $\eta$ and $\sigma$ using 250 runs of the TPE search from Optuna (Akiba et al., 2019), for each unlearning scenario. For VGG11, we use the following parameters: full-class unlearning uses $\eta = 0.0003, \sigma = 0.5$, sub-class and random both use $\eta = 0.0003, \sigma = 0.01$. For ViT, the selected parameters are: full-class $\eta = 1.5, \sigma = 0.8$, sub-class $\eta = 0.5, \sigma = 1.5$, and random $\eta = 0.01, \sigma = 0.5$. ViT and VGG use considerably different learning rates, since only a single epoch is used during the unlearning step. If minimising the runtime is a looser constraint, a smaller learning rate can be used for ViT with extra epochs of training. We stress that when selecting hyper-parameters, we selected values that yielded promising results, without rigorously fitting our results to the retrained model.

# 6 Results

## 6.1 Benchmark Evaluation

**Compute Comparison.** Figure 5 shows the runtime of JiT compared to other methods. JiT is very fast, especially in comparison to the other ZS methods, performing more than 5 times faster than Boundary Shrinking. Sufficiently short runtimes are an important desideratum of unlearning, one which JiT empirically satisfies. JiT has a computational complexity of $O(N|\mathcal{D}_f|)$, where $N$ is the number of perturbed samples and $|\mathcal{D}_f|$ is the cardinality of the forget set. Requiring only $\mathcal{D}_f$ and processing each sample just once makes JiT efficient.

**Full-class Unlearning.** We begin by comparing full class performance to that of the existing ZS methods. As seen in Tables 2 (a), 2 (b), and 1, JiT demonstrates significantly superior performance over GKT and EMMN, and is competitive with Boundary Shrinking. The authors of Chundawat et al. (2023b) note the poor scalability of both EMMN and GKT, which is evident in our results. Failing to scale to large problems or models is a significant barrier, since the value of unlearning is found mostly in large models that are expensive to train, or large datasets that are expensive to store. JiT performance is competitive with Boundary Shrinking despite having a fraction of the compute cost and, even compared to non-ZS SOTA, JiT performs reasonably; dropping only 0.6% retain set performance compared to the baseline on ViT and outperforming both UNSIR and SSD on the MIA. Our performance also holds for VGG and, when using the same hyper-parameters for the face dataset, JiT generalizes well, outperforming SCRUB, SSD, and UNSIR.

**Sub-class Unlearning.** Tables 3 (a) and 3 (b) show the performance of JiT on sub-class unlearning for ViT and VGG11. For both, JiT is able to comfortably unlearn. For ViT, it actually *over-forgets*, however this is typically an easier problem to correct, since more conservative values can always be selected. $\mathcal{D}_r$ performance also drops slightly more than usual, which a more conservative parameter set could also correct. For VGG11, however, the method is comfortably amongst the SOTA, outperforming methods that are granted access to the retained data. For ViT, JiT better minimises $\mathcal{D}_f$ compared to BDSH.

**Random Unlearning.** Tables 4 (a) and 4 (b) show method performance when forgetting 100 samples uniformly distributed across the training set. JiT is able to comfortably rival existing ZS and non-ZS SOTA methods; despite slight over-forgetting, $\mathcal{D}_r$ accuracy is almost unchanged ($\sim 1\%$ for both models). Table 5 (b) shows both ZS methods also generalize to random unlearning on ResNet18, albeit with a larger drop in retain set accuracy.

**Additional Ablations.** Table 5 (a) validates our method on a larger scale problem, with JiT achieving SOTA performance for ZS methods. As larger pretrained models can be robust to noise, we found larger

Table 2: (a) ViT Full-class unlearning performance on CIFAR-100 class Rocket. (b) VGG11 Full-class unlearning performance on CIFAR-100 class Rocket.

| | (a) | | | | | (b) | | | |
|---|---|---|---|---|---|---|---|---|---|
| METHOD | $\mathcal{D}_r$ ACC. | $\mathcal{D}_f$ ACC. | MIA | ZS | METHOD | $\mathcal{D}_r$ ACC. | $\mathcal{D}_f$ ACC. | MIA | ZS |
| BSLN | $88.9 \pm 0.0$ | $94.7 \pm 0.0$ | $94.4 \pm 0.0$ | $\times$ | BSLN | $66.3 \pm 0.0$ | $77.0 \pm 0.0$ | $97.4 \pm 0.0$ | $\times$ |
| RTRN | $90.1 \pm 0.0$ | $0.0 \pm 0.0$ | $3.2 \pm 0.5$ | $\times$ | RTRN | $63.2 \pm 0.5$ | $0.0 \pm 0.0$ | $10.4 \pm 1.1$ | $\times$ |
| FNTN | $80.8 \pm 1.4$ | $0.6 \pm 0.7$ | $19.0 \pm 8.7$ | $\times$ | FNTN | $59.7 \pm 0.4$ | $3.9 \pm 3.0$ | $13.2 \pm 4.2$ | $\times$ |
| AMN | $87.9 \pm 0.9$ | $0.0 \pm 0.0$ | $1.4 \pm 0.9$ | $\times$ | AMN | $64.3 \pm 0.4$ | $0.0 \pm 0.0$ | $1.8 \pm 0.8$ | $\times$ |
| | | | | | SCRUB | $66.2 \pm 0.1$ | $0.0 \pm 0.0$ | $8.2 \pm 1.7$ | $\times$ |
| SSD | $88.90 \pm 0.0$ | $0.0 \pm 0.0$ | $1.8 \pm 0.0$ | $\times$ | SSD | $63.79 \pm 0.0$ | $0.0 \pm 0.0$ | $8.6 \pm 0.0$ | $\times$ |
| BT | $87.5 \pm 0.5$ | $4.2 \pm 5.2$ | $0.0 \pm 0.1$ | $\times$ | BT | $65.5 \pm 0.2$ | $0.1 \pm 0.3$ | $0.0 \pm 0.1$ | $\times$ |
| UNSIR | $88.5 \pm 0.4$ | $65.3 \pm 9.1$ | $29.1 \pm 6.1$ | $\times$ | UNSIR | $64.6 \pm 0.4$ | $42.9 \pm 14.3$ | $40.7 \pm 12.1$ | $\times$ |
| GKT | $1.0 \pm 0.6$ | $0.0 \pm 0.0$ | $60.0 \pm 51.6$ | $\checkmark$ | GKT | $2.3 \pm 0.2$ | $0.0 \pm 0.0$ | $56.2 \pm 20.0$ | $\checkmark$ |
| EMMN | $84.6 \pm 0.4$ | $94.3 \pm 1.5$ | $93.7 \pm 2.2$ | $\checkmark$ | EMMN | $26.9 \pm 7.7$ | $24.3 \pm 23.7$ | $58.2 \pm 14.5$ | $\checkmark$ |
| BDSH | $87.6 \pm 0.0$ | $0.0 \pm 0.0$ | $5.0 \pm 0.0$ | $\checkmark$ | BDSH | $66.2 \pm 0.1$ | $13.0 \pm 0.0$ | $2.9 \pm 0.1$ | $\checkmark$ |
| OURS | $87.5 \pm 0.0$ | $51.9 \pm 2.13$ | $4.3 \pm 0.38$ | $\checkmark$ | OURS | $66.2 \pm 0.3$ | $14.2 \pm 0.6$ | $2.9 \pm 0.3$ | $\checkmark$ |

Table 3: (a) VGG-16 Sub-class unlearning performance on CIFAR-20 sub-class Rocket. (b) ViT Sub-class unlearning performance onCIFAR-20 sub-class Rocket

(a)

| METHOD | $\mathcal{D}_r$ ACC. | $\mathcal{D}_f$ ACC. | MIA | ZS |
|---|---|---|---|---|
| BSLN | 75.3±0.0 | 79.0±0.0 | 83.1±0.0 | × |
| RTRN | 72.9±0.2 | 11.5±2.8 | 14.1±1.3 | × |
| FNTN | 65.5±0.7 | 6.2±3.7 | 22.3±5.5 | × |
| AMN | 73.8±0.2 | 2.4±2.4 | 3.0±0.9 | × |
| SCRUB | 62.4±28.4 | 10.1±22.48 | 16.7±21.7 | × |
| SSD | 75.0±0.0 | 4.2±0.0 | 11.0±0.0 | × |
| BT | 74.9±0.2 | 48.4±16.9 | 0.1±0.1 | × |
| UNSIR | 74.1±0.2 | 57.5±10.3 | 57.4±8.6 | × |
| BDSH | 74.4±0.0 | 17.535±0.0 | 12.9±0.1 | ✓ |
| OURS | 73.7±0.8 | 19.3±18.3 | 11.2±7.8 | ✓ |

(b)

| METHOD | $D_r$ ACC. | $D_f$ ACC. | MIA | ZS |
|---|---|---|---|---|
| BSLN | 95.7 ± 0.0 | 94.5 ± 0.0 | 80.4 ± 0.0 | × |
| RTRN | 94.6 ± 0.1 | 22.3 ± 8.3 | 3.4 ± 1.1 | × |
| FNTN | 85.7 ± 3.1 | 6.2 ± 6.0 | 16.0 ± 2.7 | × |
| AMN | 93.5 ± 0.2 | 0.8 ± 1.7 | 0.8 ± 0.3 | × |
| SSD | 95.1 ± 0.0 | 5.12 ± 0.0 | 5.4 ± 0.0 | × |
| BT | 93.6 ± 0.3 | 3.3 ± 2.9 | 0.0 ± 0.1 | × |
| UNSIR | 93.3 ± 0.4 | 74.9 ± 10.1 | 27.3 ± 13.8 | × |
| BDSH | 95.7±0.0 | 48.4±0.0 | 0.1±0.0 | ✓ |
| OURS | 92.2±0.0 | 0.0±0.0 | 14.66±8.8 | ✓ |

perturbations were required to induce forgetting. To keep the input in-domain, we apply normalization to the noised image via: $\frac{(x+\xi)}{\sqrt{(1+\sigma^2)}}$. Finally, we take the 50 samples from class Rocket for which the MIA model outputs the highest probability. In other words, the 50 samples that are the "worst-case scenario" and which leak the most information about the training set. Table 6 shows the performance of each method when attempting to unlearn this subset. JiT performs very well here, performing comparatively with all other methods implemented. Interestingly, both SCRUB and BDSH struggle with forgetting the worst-case samples.

## 7    Discussion

JiT is by far the fastest ZS unlearning method benchmarked, a critical characteristic for satisfying the unlearning task. JiT is competitive with state-of-the-art performance in the ZS unlearning domain, as well as competing with non-ZS methods in the sub-class and random unlearning tasks despite their easier task. The entropy experiments highlight that JiT is able to replicate the output entropy of a retrained model over a forget set, while preserving retain set performance. When compared to existing ZS methods, JiT can be considered a strong baseline. It is fast and performant, and performed acceptably across all benchmarks implemented. If time constraints are ignored, BDSH is more stable and less sensitive to hyper-parameter selection, on account of taking the true gradients of the model with respect to the input, rather than the approximation we employ with JiT. However, in practice the poor time complexity of BDSH will likely

Table 4: (a) VGG11 Random unlearning performance for 100 samples from CIFAR-10. (b)ViT Random unlearning performance for 100 samples from CIFAR-10.

(a)

| METHOD | $\mathcal{D}_r$ ACC. | $\mathcal{D}_f$ ACC. | MIA | ZS |
|---|---|---|---|---|
| BSLN | 87.0±0.0 | 92.0±3.6 | 70.1±5.4 | × |
| RTRN | 87.7±0.2 | 91.0±2.5 | 78.9±3.5 | × |
| FNTN | 84.4±0.8 | 86.4±4.4 | 70.8±4.7 | × |
| AMN | 86.8±0.3 | 51.3±4.4 | 13.1±2.9 | × |
| SCRUB | 87.7±0.1 | 92.7±2.9 | 71.8±5.2 | × |
| SSD | 85.6±2.7 | 90.8±3.7 | 66.7±5.9 | × |
| BT | 86.9±0.2 | 82.5±4.9 | 40.8±6.3 | × |
| BDSH | 86.9±0.1 | 92.2±3.4 | 69.8±5.1 | ✓ |
| OURS | 86.3±0.3 | 88.7±3.9 | 64.2±5.2 | ✓ |

(b)

| METHOD | $\mathcal{D}_r$ ACC. | $\mathcal{D}_f$ ACC. | MIA | ZS |
|---|---|---|---|---|
| BSLN | 98.9 ± 0.0 | 100.0 ± 0.0 | 90.8 ± 3.5 | × |
| RTRN | 98.6 ± 0.1 | 98.8 ± 0.8 | 91.8 ± 1.8 | × |
| FNTN | 97.3 ± 0.3 | 97.2 ± 1.0 | 86.1 ± 2.1 | × |
| AMN | 97.6 ± 0.3 | 73.5 ± 5.1 | 10.4 ± 4.9 | × |
| SSD | 98.0 ± 1.6 | 98.1 ± 2.4 | 85.5 ± 0.1 | × |
| BT | 97.6 ± 0.4 | 86.7 ± 3.6 | 33.5 ± 5.6 | × |
| BDSH | 98.0±0.29 | 97.9±1.6 | 78.8±0.0 | ✓ |
| OURS | 98.0±0.3 | 98.0±1.5 | 78.8±4.0 | ✓ |

Table 5: (a) Zero-shot methods performance on a ViT-L trained on ILSVRC Imagenet. (b) ResNet18 random unlearning performance on CIFAR10.

(a)

| Method | $\mathcal{D}_r$ Acc. | $\mathcal{D}_f$ Acc | MIA |
|---|---|---|---|
| BSLN | 86.0 | 100.0 | 94.0 |
| BDSH | 85.9 | 60.0 | 0.0 |
| OURS | 83.6 | 10.0 | 0.0 |

(b)

| Method | $\mathcal{D}_r$ Acc. | $\mathcal{D}_f$ Acc | MIA |
|---|---|---|---|
| BSLN | 90.7 | 95.3 | 75.8 |
| RTRN | 91.5 | 94.1 | 74.2 |
| FNTN | 88.0 | 90.0 | 74.6 |
| AMN | 90.2 | 59.0 | 25.2 |
| SSD | 88.7 | 93.6 | 72.7 |
| BT | 90.2 | 90.0 | 49.3 |
| BDSH | 89.5 | 91.6 | 71.1 |
| OURS | 85.6 | 92.8 | 68.7 |

prove prohibitive when trying to unlearn from internet-scale models, whereas JiT is amongst the fastest methods we benchmarked. Future work could explore the efficacy of using JiT with exact gradients, or a more specialised gradient approximation. JiT has the potential for positive societal impacts, aiding the preservation of individual privacy. However, due to a lack of certification, poor use of JiT could result in organizations believing they have removed the influence of an individual's data when they haven't.

## 8 Limitations

The appendix provides a sensitivity analysis (10.2). It demonstrates that JiT can be sensitive to hyperparameter selection. This is a by-product of having no access to $\mathcal{D}_r$ to finetune, and also due to our gradient approximation method. Exact gradients are slower, but may prove more stable. Importantly, we note that better stability can be achieved through gradient clipping, though this came at the cost of method performance.

Since we minimise the gradient over each forget sample independently, we advise caution when using a model with batch normalization; since this changes the model's mapping of single input/output to batch input/output (Gulrajani et al., 2017). This limitation can be mitigated by selecting an alternative choice of model or normalization strategy (e.g. layer norms).

Table 6: Method performance when evaluated on a VGG11 against the 50 worst-case samples in class Rocket for CIFAR100.

| Method | $\mathcal{D}_r$ Acc. | $\mathcal{D}_f$ Acc | MIA |
|---|---|---|---|
| BSLN | 66.6 | 77.0 | 100.0 |
| RTRN | 64.5 | 0.0 | 0.0 |
| FNTN | 59.7 | 0.0 | 6.0 |
| AMN | 65.0 | 0.0 | 4.0 |
| SSD | 65.9 | 8.0 | 0.0 |
| BT | 66.0 | 0.0 | 0.0 |
| SCRUB | 66.5 | 34.0 | 0.0 |
| BDSH | 66.7 | 65 | 100.0 |
| OURS | 64.5 | 0.0 | 2.0 |

JiT, like most SOTA methods, is not certified. The impact of this will vary by application domain, but may preclude its use in especially sensitive areas. Finally, we note that our method is specifically tailored to classification tasks. While it is possible a variation of this approach could work for large generative models, we restrict our focus and our claims to larger classifiers and leave generative applications to future work.

## 9 Conclusion

Unlearning is an important, challenging problem. The ZS setting is amongst the hardest, requiring delicate treatment of the unlearning process to ensure model performance is protected. In this work, we approached this challenge from an information theoretic perspective, deriving an unlearning algorithm directly from the notion of minimising information gained from a sample. We demonstrate empirically the geometric insights behind why JiT can effectively tackle the ZS unlearning problem, alongside showing experimentally that JiT can reconstruct behaviour analagous to that of a model retrained from scratch. JiT achieves performance competitive with state-of-the-art ZS and non-ZS methods. We evaluate JiT on a range of benchmarks, demonstrating its efficacy in full-class, sub-class, and random unlearning, across multiple models. Future work is needed to establish a stronger theoretical relationship between forgetting and information theory, as well as exploring whether this can be formalized to provide guarantees on forgetting using information theoretic approaches.

**Acknowledgments**

This work was supported by EPSRC CDT AgriFoRwArdS [grant number EP/S023917/1], and EPSRC DTP [grant number EP/W524633/1].

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

## 10 Appendix

### 10.1 Method Algorithm

---

**Algorithm 1** JıT UNLEARNING

---

**INPUT**: The trained model $f_\theta(\cdot)$ and the forget set $\mathcal{S}$.
**PARAMETER**: $\eta, \sigma, N$
**OUTPUT**: $f_{\hat{\theta}}(\cdot) = \mathcal{U}_\mathcal{S}(f_\theta(\cdot))$

1: Initialise optim$(\theta, lr = \eta)$
2: **for** $x$ in $\mathcal{S}$ **do**
3:    $\ell = 0$
4:    **for** $i$ in range$(N)$ **do**
5:       $x' = x + \boldsymbol{\xi}$ for $\boldsymbol{\xi} \sim \mathcal{N}(0, \sigma^2)$
6:       $k = \frac{\left\| f_\theta(\boldsymbol{x}) - f_\theta(\boldsymbol{x}') \right\|_2}{\|\boldsymbol{\xi}\|_2}$
7:       $\ell = \ell + k$
8:    **end for**
9: **end for**
10: $\ell = \ell/N$
11: $\hat{\theta} \leftarrow$ optim$\{\nabla_\theta \ell\}$
12: **return** $f_{\hat{\theta}}(\cdot)$

---

Figure 6: Pseudocode algorithm for JiT Unlearning

## 10.2 Sensitivity Analysis

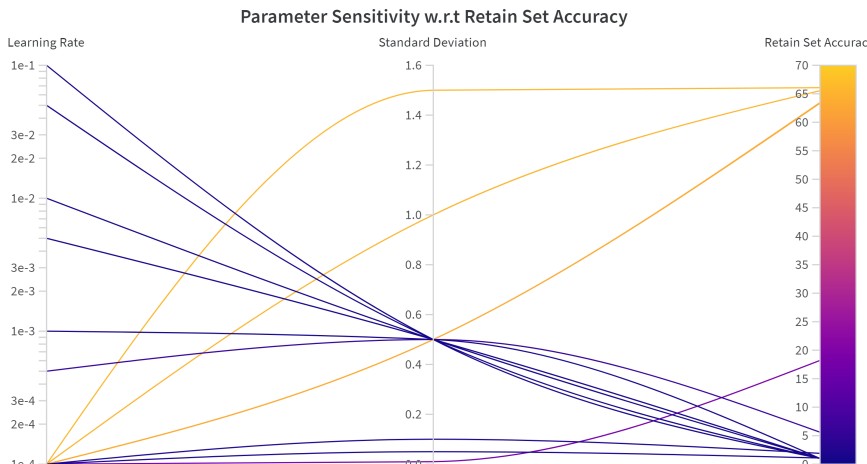

Figure 7: Plot of the $\mathcal{D}_r$ sensitivity to change in hyper-parameters for VGG11 full-class forgetting.

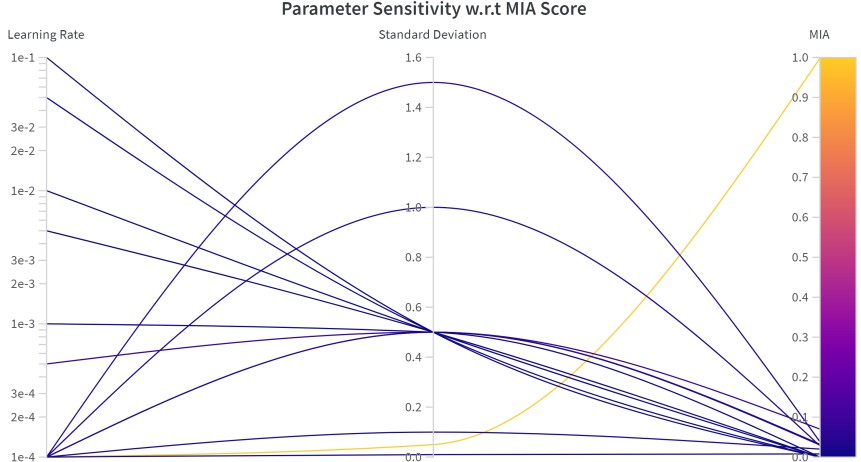

Figure 8: Plot of the MIA sensitivity to change in hyper-parameters for VGG11 full-class forgetting.

Figures 7 and 8 show the sensitivity of key metrics to changes in the hyper-parameters $\eta$ and $\sigma$. In general, the approach is robust to small perturbations of the learning rate, but naturally as it increases by orders of magnitude, performance varies significantly. For $\sigma$, in general increasing the noise actually reduced the forgetting/increased $\mathcal{D}_r$ accuracy. This is because VGG was so sensitivity to noise, and so small values of $\sigma$ did little to reduce the divergence of model output, and thus for $\sigma < 1$ the loss increases due to the division in the loss term. For models that are robust to additive noise, the relationship between $\sigma$ and performance is often parabolic.

Choice of how many perturbed variants is simple. The more samples the more stable the forgetting, and the only trade-off is compute time.

### 10.3 Class Breakdown of Method Performance

Below are the forget-class breakdowns of each method's performance for full-class and sub-class unlearning. For each unlearning scenario, the same parameters are used across all classes; this can lead to significant variance in method performance.

Table 7: ViT CIFAR-100 full-class unlearning breakdown

| Method | Forget Class | $\mathcal{D}_r$ | $\mathcal{D}_f$ | MIA | Method Runtime | ZS |
|---|---|---|---|---|---|---|
| Baseline | baby | $88.9 \pm 0.0$ | $90.2 \pm 0.0$ | $75.6 \pm 0.0$ | $459.7 \pm 83.0$ | ✗ |
| Baseline | lamp | $88.8 \pm 0.0$ | $97.2 \pm 0.0$ | $95.6 \pm 0.0$ | $504.1 \pm 131.7$ | ✗ |
| Baseline | mushroom | $88.9 \pm 0.0$ | $94.9 \pm 0.0$ | $92.8 \pm 0.0$ | $474.6 \pm 60.2$ | ✗ |
| Baseline | rocket | $88.9 \pm 0.0$ | $94.7 \pm 0.0$ | $94.4 \pm 0.0$ | $470.3 \pm 85.7$ | ✗ |
| Baseline | sea | $88.9 \pm 0.0$ | $90.5 \pm 0.0$ | $80.4 \pm 0.0$ | $549.3 \pm 81.8$ | ✗ |
| Retrain | baby | $90.3 \pm 0.1$ | $0.0 \pm 0.0$ | $21.5 \pm 2.8$ | $2029.1 \pm 169.7$ | ✗ |
| Retrain | lamp | $90.1 \pm 0.2$ | $0.0 \pm 0.0$ | $2.3 \pm 0.7$ | $2115.0 \pm 196.3$ | ✗ |
| Retrain | mushroom | $90.0 \pm 0.2$ | $0.0 \pm 0.0$ | $0.7 \pm 0.4$ | $1948.8 \pm 123.1$ | ✗ |
| Retrain | rocket | $90.1 \pm 0.1$ | $0.0 \pm 0.0$ | $3.2 \pm 0.5$ | $1939.7 \pm 109.7$ | ✗ |
| Retrain | sea | $90.3 \pm 0.2$ | $0.0 \pm 0.0$ | $8.4 \pm 2.2$ | $2176.1 \pm 92.8$ | ✗ |
| Finetune | baby | $80.7 \pm 1.4$ | $0.0 \pm 0.0$ | $26.8 \pm 12.7$ | $1461.8 \pm 96.2$ | ✗ |
| Finetune | lamp | $80.2 \pm 1.5$ | $0.4 \pm 0.9$ | $11.8 \pm 4.3$ | $1537.4 \pm 133.2$ | ✗ |
| Finetune | mushroom | $81.1 \pm 0.8$ | $2.3 \pm 2.4$ | $7.1 \pm 1.9$ | $1421.7 \pm 80.5$ | ✗ |
| Finetune | rocket | $80.8 \pm 1.4$ | $0.5 \pm 0.7$ | $19.0 \pm 8.7$ | $1404.9 \pm 82.8$ | ✗ |
| Finetune | sea | $80.8 \pm 1.4$ | $0.0 \pm 0.0$ | $22.0 \pm 7.1$ | $1525.0 \pm 147.3$ | ✗ |
| Amnesiac | baby | $88.4 \pm 0.7$ | $0.0 \pm 0.0$ | $1.8 \pm 0.3$ | $1132.0 \pm 147.1$ | ✗ |
| Amnesiac | lamp | $88.4 \pm 0.6$ | $0.0 \pm 0.0$ | $2.7 \pm 0.4$ | $1146.5 \pm 97.5$ | ✗ |
| Amnesiac | mushroom | $88.3 \pm 0.7$ | $0.0 \pm 0.0$ | $0.5 \pm 0.2$ | $1071.9 \pm 98.1$ | ✗ |
| Amnesiac | rocket | $87.9 \pm 0.9$ | $0.0 \pm 0.0$ | $1.0 \pm 0.6$ | $1029.7 \pm 70.8$ | ✗ |
| Amnesiac | sea | $88.3 \pm 0.3$ | $0.0 \pm 0.0$ | $0.8 \pm 0.2$ | $1176.2 \pm 136.7$ | ✗ |
| SSD | baby | $88.6 \pm 0.0$ | $0.0 \pm 0.0$ | $0.6 \pm 0.0$ | $685.40 \pm 97.4$ | ✗ |
| SSD | lamp | $89.1 \pm 0.0$ | $36.9 \pm 0.0$ | $0.4 \pm 0.0$ | $741.23 \pm 170.6$ | ✗ |
| SSD | mushroom | $88.8 \pm 0.0$ | $0.0 \pm 0.0$ | $3.8 \pm 0.0$ | $650.93 \pm 76.4$ | ✗ |
| SSD | rocket | $88.9 \pm 0.0$ | $0.0 \pm 0.0$ | $1.8 \pm 0.0$ | $655.64 \pm 65.4$ | ✗ |
| SSD | sea | $88.0 \pm 0.0$ | $0.0 \pm 0.0$ | $3.2 \pm 0.0$ | $767.35 \pm 177.8$ | ✗ |
| Teacher | baby | $87.5 \pm 0.4$ | $23.8 \pm 22.5$ | $0.0 \pm 0.0$ | $610.0 \pm 98.1$ | ✗ |
| Teacher | lamp | $87.5 \pm 0.4$ | $25.2 \pm 12.5$ | $0.1 \pm 0.2$ | $668.6 \pm 135.1$ | ✗ |
| Teacher | mushroom | $87.4 \pm 0.4$ | $12.8 \pm 5.9$ | $0.0 \pm 0.1$ | $602.9 \pm 33.2$ | ✗ |
| Teacher | rocket | $87.5 \pm 0.5$ | $4.2 \pm 5.2$ | $0.0 \pm 0.1$ | $602.6 \pm 63.2$ | ✗ |
| Teacher | sea | $87.7 \pm 0.2$ | $51.1 \pm 17.4$ | $0.0 \pm 0.0$ | $661.8 \pm 117.1$ | ✗ |
| UNSIR | baby | $88.8 \pm 0.4$ | $2.0 \pm 1.2$ | $14.3 \pm 6.1$ | $954.6 \pm 110.3$ | ✗ |
| UNSIR | lamp | $88.5 \pm 0.4$ | $70.9 \pm 4.4$ | $29.4 \pm 4.8$ | $1002.2 \pm 135.1$ | ✗ |
| UNSIR | mushroom | $88.4 \pm 0.6$ | $83.9 \pm 2.9$ | $21.3 \pm 2.7$ | $891.8 \pm 69.9$ | ✗ |
| UNSIR | rocket | $88.5 \pm 0.4$ | $65.3 \pm 9.1$ | $29.1 \pm 6.1$ | $868.8 \pm 47.9$ | ✗ |
| UNSIR | sea | $88.8 \pm 0.2$ | $13.9 \pm 6.2$ | $9.1 \pm 4.7$ | $986.1 \pm 149.6$ | ✗ |
| GKT | baby | $1.0 \pm 0.1$ | $0.0 \pm 0.0$ | $70.0 \pm 48.3$ | $1074.9 \pm 791.5$ | ✓ |
| GKT | lamp | $1.0 \pm 0.1$ | $0.0 \pm 0.0$ | $60.0 \pm 51.6$ | $1924.5 \pm 1269.6$ | ✓ |
| GKT | mushroom | $1.0 \pm 0.1$ | $0.0 \pm 0.0$ | $60.0 \pm 51.6$ | $1793.3 \pm 1100.7$ | ✓ |
| GKT | rocket | $1.0 \pm 0.1$ | $0.0 \pm 0.0$ | $60.0 \pm 51.6$ | $1944.0 \pm 1227.4$ | ✓ |

**Table 7 (Continued)**

| Method | Forget Class | $\mathcal{D}_r$ | $\mathcal{D}_f$ | MIA | Method Runtime | ZS |
|---|---|---|---|---|---|---|
| GKT | sea | $1. \pm 0.1$ | $0.0 \pm 0.00$ | $70.0 \pm 48.3$ | $1788.3 \pm 1150.9$ | ✓ |
| EMMN | baby | $84.6 \pm 0.3$ | $87.9 \pm 3.0$ | $74.8 \pm 6.5$ | $1064.7 \pm 202.6$ | ✓ |
| EMMN | lamp | $85.0 \pm 0.4$ | $93.0 \pm 2.0$ | $87.6 \pm 3.0$ | $1100.8 \pm 212.1$ | ✓ |
| EMMN | mushroom | $84.7 \pm 0.3$ | $93.1 \pm 1.8$ | $91.1 \pm 2.2$ | $1068.9 \pm 132.3$ | ✓ |
| EMMN | rocket | $84.6 \pm 0.4$ | $94.3 \pm 1.5$ | $93.7 \pm 2.2$ | $1115.3 \pm 179.3$ | ✓ |
| EMMN | sea | $84.7 \pm 0.3$ | $89.5 \pm 4.3$ | $74.6 \pm 7.4$ | $1075.8 \pm 216.6$ | ✓ |
| BDSH | baby | $83.9 \pm 0.0$ | $0.0 \pm 0.0$ | $8.6 \pm 0.0$ | $1388.1 \pm 310.5$ | ✓ |
| BDSH | lamp | $86.9 \pm 0.0$ | $2.0 \pm 0.0$ | $28.4 \pm 0.0$ | $1386.4 \pm 304.9$ | ✓ |
| BDSH | mushroom | $87.8 \pm 0.0$ | $0.0 \pm 0.0$ | $0.8 \pm 0.0$ | $1398.2 \pm 305.3$ | ✓ |
| BDSH | rocket | $87.6 \pm 0.0$ | $0.0 \pm 0.0$ | $5.0 \pm 0.0$ | $1396.9 \pm 306.4$ | ✓ |
| BDSAH | sea | $87.3 \pm 0.0$ | $3.0 \pm 0.0$ | $0.8 \pm 0.0$ | $1414.4 \pm 146.2$ | ✓ |
| Ours | baby | $87.2 \pm 0.0$ | $38.7 \pm 1.0$ | $0.4 \pm 0.0$ | $607.9 \pm 130.3$ | ✓ |
| Ours | lamp | $88.4 \pm 0.0$ | $93.9 \pm 0.3$ | $49.8 \pm 2.1$ | $597.0 \pm 143.6$ | ✓ |
| Ours | mushroom | $87.7 \pm 0.0$ | $77.9 \pm 0.6$ | $4.8 \pm 0.4$ | $656.2 \pm 141.6$ | ✓ |
| Ours | rocket | $87.5 \pm 0.0$ | $51.9 \pm 2.1$ | $4.3 \pm 0.4$ | $629.4 \pm 116.3$ | ✓ |
| Ours | sea | $83.8 \pm 0.1$ | $24.0 \pm 1.2$ | $16.3 \pm 0.4$ | $596.3 \pm 151.9$ | ✓ |

Table 8: ViT CIFAR-20 sub-class unlearning breakdown

| Method | Forget Class | $\mathcal{D}_r$ | $\mathcal{D}_f$ | MIA | Method Runtime | ZS |
|---|---|---|---|---|---|---|
| Baseline | baby | $95.7 \pm 0.0$ | $96.4 \pm 0.0$ | $91.6 \pm 0.0$ | $443.0 \pm 48.7$ | ✗ |
| Baseline | lamp | $95.8 \pm 0.0$ | $89.6 \pm 0.0$ | $81.0 \pm 0.0$ | $419.7 \pm 25.2$ | ✗ |
| Baseline | mushroom | $95.7 \pm 0.0$ | $97.0 \pm 0.0$ | $77.8 \pm 0.0$ | $525.7 \pm 124.3$ | ✗ |
| Baseline | rocket | $95.7 \pm 0.0$ | $94.5 \pm 0.0$ | $80.4 \pm 0.0$ | $535.0 \pm 75.3$ | ✗ |
| Baseline | sea | $95.7 \pm 0.0$ | $99.2 \pm 0.0$ | $88.4 \pm 0.0$ | $475.5 \pm 120.3$ | ✗ |
| Retrain | baby | $94.5 \pm 0.2$ | $93.2 \pm 1.1$ | $77.4 \pm 3.4$ | $2148.7 \pm 112.5$ | ✗ |
| Retrain | lamp | $94.7 \pm 0.1$ | $34.5 \pm 8.6$ | $5.6 \pm 1.6$ | $2149.3 \pm 124.4$ | ✗ |
| Retrain | mushroom | $94.6 \pm 0.1$ | $26.6 \pm 6.4$ | $2.3 \pm 0.5$ | $2161.5 \pm 81.6$ | ✗ |
| Retrain | rocket | $94.6 \pm 0.1$ | $22.3 \pm 8.3$ | $3.4 \pm 1.1$ | $2168.3 \pm 118.4$ | ✗ |
| Retrain | sea | $94.6 \pm 0.2$ | $95.1 \pm 0.8$ | $66.0 \pm 3.8$ | $2142.0 \pm 85.9$ | ✗ |
| Finetune | baby | $87.6 \pm 0.8$ | $85.4 \pm 4.5$ | $66.6 \pm 7.1$ | $1426.9 \pm 72.7$ | ✗ |
| Finetune | lamp | $87.7 \pm 0.5$ | $16.9 \pm 10.4$ | $14.7 \pm 3.9$ | $1424.9 \pm 71.4$ | ✗ |
| Finetune | mushroom | $87.4 \pm 0.9$ | $15.7 \pm 12.1$ | $9.2 \pm 4.1$ | $1442.4 \pm 113.2$ | ✗ |
| Finetune | rocket | $85.7 \pm 3.1$ | $6.2 \pm 6.0$ | $16.0 \pm 2.7$ | $1436.6 \pm 117.9$ | ✗ |
| Finetune | sea | $87.6 \pm 1.6$ | $89.2 \pm 4.2$ | $65.0 \pm 12.9$ | $1512.9 \pm 129.8$ | ✗ |
| Amnesiac | baby | $93.3 \pm 0.3$ | $38.8 \pm 7.4$ | $0.9 \pm 0.7$ | $1025.0 \pm 35.2$ | ✗ |
| Amnesiac | lamp | $93.7 \pm 0.5$ | $0.6 \pm 1.5$ | $2.0 \pm 1.0$ | $1009.0 \pm 37.7$ | ✗ |
| Amnesiac | mushroom | $93.4 \pm 0.5$ | $0.2 \pm 0.4$ | $1.5 \pm 0.5$ | $1131.8 \pm 137.6$ | ✗ |
| Amnesiac | rocket | $93.5 \pm 0.2$ | $0.8 \pm 1.7$ | $0.8 \pm 0.3$ | $1186.6 \pm 107.5$ | ✗ |
| Amnesiac | sea | $93.3 \pm 0.2$ | $21.4 \pm 8.5$ | $0.4 \pm 0.2$ | $1070.4 \pm 138.1$ | ✗ |
| SSD | baby | $95.5 \pm 0.0$ | $94.1 \pm 0.0$ | $77.2 \pm 0.0$ | $736.0 \pm 12.0$ | ✗ |
| SSD | lamp | $95.5 \pm 0.0$ | $14.6 \pm 0.0$ | $3.2 \pm 0.0$ | $729.0 \pm 73.1$ | ✗ |
| SSD | mushroom | $95.5 \pm 0.0$ | $6.7 \pm 0.0$ | $0.4 \pm 0.0$ | $718.8 \pm 73.4$ | ✗ |

**Table 8 (Continued)**

| Method | Forget Class | $\mathcal{D}_r$ | $\mathcal{D}_f$ | MIA | Method Runtime | ZS |
|---|---|---|---|---|---|---|
| SSD | rocket | 95.1 ± 0.0 | 5.1 ± 0.0 | 5.4 ± 0.0 | 699.3 ± 72.5 | × |
| SSD | sea | 95.6 ± 0.0 | 97.1 ± 0.0 | 82.2 ± 0.0 | 645.7 ± 53.4 | × |
| Teacher | baby | 93.0 ± 0.5 | 46.7 ± 17.9 | 0.0 ± 0.1 | 553.9 ± 65.9 | × |
| Teacher | lamp | 93.6 ± 0.7 | 8.2 ± 7.1 | 0.1 ± 0.2 | 558.1 ± 63.9 | × |
| Teacher | mushroom | 93.6 ± 0.4 | 13.0 ± 9.1 | 0.0 ± 0.0 | 620.1 ± 111.7 | × |
| Teacher | rocket | 93.6 ± 0.3 | 3.3 ± 2.9 | 0.0 ± 0.1 | 631.9 ± 115.2 | × |
| Teacher | sea | 93.6 ± 0.3 | 26.0 ± 14.0 | 0.2 ± 0.1 | 586.7 ± 89.7 | × |
| UNSIR | baby | 93.2 ± 0.3 | 94.5 ± 0.8 | 88.0 ± 3.1 | 871.1 ± 60.7 | × |
| UNSIR | lamp | 93.4 ± 0.5 | 76.5 ± 5.2 | 36.5 ± 11.7 | 899.8 ± 72.0 | × |
| UNSIR | mushroom | 93.1 ± 0.6 | 79.8 ± 7.6 | 19.0 ± 7.4 | 925.7 ± 117.5 | × |
| UNSIR | rocket | 93.3 ± 0.4 | 74.9 ± 10.1 | 27.3 ± 13.8 | 983.1 ± 143.2 | × |
| UNSIR | sea | 93.3 ± 0.3 | 94.3 ± 2.3 | 77.0 ± 7.2 | 1024.5 ± 144.8 | × |
| BDSH | baby | 95.4 ± 0.0 | 93.3 ± 0.0 | 18.8 ± 0.0 | 1163 ± 49.8 | ✓ |
| BDSH | lamp | 95.8 ± 0.0 | 89.6 ± 0.0 | 80.8 ± 0.0 | 1152.5 ± 53.9 | ✓ |
| BDSH | mushroom | 95.7 ± 0.0 | 88.4 ± 0.0 | 2.6 ± 0.0 | 1087.9 ± 200.2 | ✓ |
| BDSH | rocket | 95.7 ± 0.0 | 48.4 ± 0.0 | 1.4 ± 0.0 | 1087.2 ± 212.6 | ✓ |
| BDSH | sea | 95.1 ± 0.0 | 78.9 ± 0.0 | 4.6 ± 0.0 | 1101.9 ± 215.9 | ✓ |
| Ours | baby | 87.4 ± 0.0 | 0.0 ± 0.0 | 0.8 ± 0.00 | 532.2 ± 112.5 | ✓ |
| Ours | lamp | 90.2 ± 0.0 | 0.0 ± 0.0 | 22.9 ± 0.2 | 560.0 ± 58.5 | ✓ |
| Ours | mushroom | 93.7 ± 0.0 | 0.0 ± 0.0 | 1.4 ± 0.0 | 563.8 ± 69.1 | ✓ |
| Ours | rocket | 92.2 ± 0.0 | 0.0 ± 0.0 | 14.7 ± 0.1 | 535.1 ± 105.7 | ✓ |
| Ours | sea | 87.4 ± 0.0 | 0.0 ± 0.0 | 3.4 ± 0.0 | 542.2 ± 87.8 | ✓ |

Table 9: VGG11 CIFAR-100 class unlearning breakdown

| Method | Forget Class | $\mathcal{D}_r$ | $\mathcal{D}_f$ | MIA | Method Runtime | ZS |
|---|---|---|---|---|---|---|
| Baseline | baby | 66.9 ± 0.0 | 52.0 ± 0.0 | 88.2 ± 0.0 | 74.9 ± 0.6 | × |
| Baseline | lamp | 66.8 ± 0.0 | 61.0 ± 0.0 | 96.8 ± 0.0 | 75.1 ± 0.9 | × |
| Baseline | mushroom | 66.7 ± 0.0 | 73.0 ± 0.0 | 97.4 ± 0.0 | 74.1 ± 1.1 | × |
| Baseline | rocket | 66.6 ± 0.0 | 77.0 ± 0.0 | 97.4 ± 0.0 | 75.0 ± 1.0 | × |
| Baseline | sea | 66.7 ± 0.0 | 75.0 ± 0.0 | 82.6 ± 0.0 | 75.5 ± 1.1 | × |
| Retrain | baby | 63.5 ± 0.5 | 0.0 ± 0.0 | 12.2 ± 1.9 | 1100.5 ± 2.6 | × |
| Retrain | lamp | 63.4 ± 0.5 | 0.0 ± 0.0 | 9.4 ± 1.2 | 1099.4 ± 2.1 | × |
| Retrain | mushroom | 63.5 ± 0.3 | 0.0 ± 0.0 | 6.6 ± 1.0 | 1100.0 ± 2.1 | × |
| Retrain | rocket | 63.2 ± 0.5 | 0.0 ± 0.0 | 10.4 ± 1.1 | 1100.0 ± 3.0 | × |
| Retrain | sea | 63.3 ± 0.3 | 0.0 ± 0.0 | 18.4 ± 1.0 | 1100.3 ± 2.4 | × |
| Finetune | baby | 60.0 ± 0.6 | 0.0 ± 0.0 | 26.5 ± 5.1 | 102.8 ± 0.7 | × |
| Finetune | lamp | 60.0 ± 0.6 | 1.8 ± 1.2 | 18.4 ± 3.2 | 102.8 ± 1.5 | × |
| Finetune | mushroom | 60.4 ± 0.6 | 1.4 ± 1.3 | 12.9 ± 3.0 | 102.4 ± 1.2 | × |
| Finetune | rocket | 59.7 ± 0.4 | 3.9 ± 3.0 | 13.2 ± 4.2 | 103.2 ± 1.4 | × |
| Finetune | sea | 59.7 ± 0.6 | 0.0 ± 0.0 | 28.0 ± 7.1 | 103.4 ± 0.6 | × |
| Amnesiac | baby | 64.7 ± 0.4 | 0.0 ± 0.0 | 4.3 ± 1.1 | 99.5 ± 3.4 | × |

**Table 9 (Continued)**

| Method | Forget Class | $\mathcal{D}_r$ | $\mathcal{D}_f$ | MIA | Method Runtime | ZS |
|---|---|---|---|---|---|---|
| Amnesiac | lamp | $64.6 \pm 0.4$ | $0.0 \pm 0.0$ | $4.8 \pm 1.2$ | $99.9 \pm 3.0$ | × |
| Amnesiac | mushroom | $64.5 \pm 0.5$ | $0.0 \pm 0.0$ | $3.1 \pm 1.1$ | $100.8 \pm 3.3$ | × |
| Amnesiac | rocket | $64.3 \pm 0.4$ | $0.0 \pm 0.0$ | $1.8 \pm 0.8$ | $99.7 \pm 2.8$ | × |
| Amnesiac | sea | $64.4 \pm 0.4$ | $0.0 \pm 0.0$ | $1.2 \pm 0.3$ | $100.2 \pm 3.5$ | × |
| SCRUB | baby | $67.1 \pm 0.1$ | $0.0 \pm 0.0$ | $9.3 \pm 1.5$ | $109.1 \pm 2.6$ | × |
| SCRUB | lamp | $66.8 \pm 0.1$ | $0.0 \pm 0.0$ | $8.7 \pm 0.9$ | $108.5 \pm 3.5$ | × |
| SCRUB | mushroom | $67.1 \pm 0.1$ | $0.0 \pm 0.0$ | $9.4 \pm 0.5$ | $107.6 \pm 2.8$ | × |
| SCRUB | rocket | $66.2 \pm 0.1$ | $0.0 \pm 0.0$ | $8.2 \pm 1.7$ | $108.7 \pm 2.5$ | × |
| SCRUB | sea | $66.7 \pm 0.1$ | $0.0 \pm 0.0$ | $6.3 \pm 2.4$ | $107.7 \pm 2.7$ | × |
| SSD | baby | $52.7 \pm 0.0$ | $0.0 \pm 0.0$ | $7.4 \pm 0.0$ | $81.3 \pm 1.3$ | × |
| SSD | lamp | $65.4 \pm 0.0$ | $0.0 \pm 0.0$ | $5.8 \pm 0.0$ | $81.6 \pm 0.8$ | × |
| SSD | mushroom | $62.2 \pm 0.0$ | $0.0 \pm 0.0$ | $14.6 \pm 0.0$ | $81.2 \pm 1.0$ | × |
| SSD | rocket | $63.8 \pm 0.0$ | $0.0 \pm 0.0$ | $8.6 \pm 0.0$ | $81.5 \pm 0.9$ | × |
| SSD | sea | $32.8 \pm 0.0$ | $0.0 \pm 0.0$ | $7.0 \pm 0.0$ | $81.3 \pm 1.2$ | × |
| Teacher | baby | $66.1 \pm 0.3$ | $1.0 \pm 1.1$ | $0.1 \pm 0.1$ | $79.6 \pm 0.8$ | × |
| Teacher | lamp | $66.0 \pm 0.2$ | $0.4 \pm 1.0$ | $0.0 \pm 0.0$ | $79.1 \pm 1.3$ | × |
| Teacher | mushroom | $65.7 \pm 0.3$ | $0.9 \pm 1.5$ | $0.1 \pm 0.1$ | $79.6 \pm 1.4$ | × |
| Teacher | rocket | $65.5 \pm 0.2$ | $0.1 \pm 0.3$ | $0.0 \pm 0.1$ | $79.1 \pm 1.0$ | × |
| Teacher | sea | $65.6 \pm 0.2$ | $5.2 \pm 4.6$ | $0.0 \pm 0.0$ | $86.7 \pm 24.0$ | × |
| UNSIR | baby | $64.9 \pm 0.4$ | $3.8 \pm 2.3$ | $29.4 \pm 11.0$ | $111.1 \pm 2.0$ | × |
| UNSIR | lamp | $64.7 \pm 0.4$ | $25.9 \pm 6.1$ | $17.0 \pm 5.9$ | $111.6 \pm 2.5$ | × |
| UNSIR | mushroom | $64.8 \pm 0.3$ | $21.1 \pm 10.7$ | $11.3 \pm 5.2$ | $111.3 \pm 2.9$ | × |
| UNSIR | rocket | $64.6 \pm 0.4$ | $42.9 \pm 14.3$ | $40.7 \pm 12.1$ | $112.1 \pm 3.5$ | × |
| UNSIR | sea | $64.5 \pm 0.3$ | $13.9 \pm 7.5$ | $22.9 \pm 4.5$ | $111.3 \pm 2.9$ | × |
| GKT | baby | $2.3 \pm 0.3$ | $0.0 \pm 0.0$ | $47.9 \pm 26.5$ | $634.3 \pm 9.2$ | ✓ |
| GKT | lamp | $2.4 \pm 0.4$ | $0.0 \pm 0.0$ | $45.3 \pm 23.7$ | $630.8 \pm 8.6$ | ✓ |
| GKT | mushroom | $2.2 \pm 0.3$ | $0.0 \pm 0.0$ | $47.4 \pm 11.8$ | $628.9 \pm 6.0$ | ✓ |
| GKT | rocket | $2.3 \pm 0.2$ | $0.0 \pm 0.0$ | $56.2 \pm 20.0$ | $629.0 \pm 6.9$ | ✓ |
| GKT | sea | $2.4 \pm 0.4$ | $0.0 \pm 0.0$ | $60.5 \pm 28.6$ | $632.0 \pm 5.6$ | ✓ |
| EMMN | baby | $30.1 \pm 8.6$ | $7.8 \pm 9.2$ | $54.0 \pm 11.2$ | $274.7 \pm 4.4$ | ✓ |
| EMMN | lamp | $33.0 \pm 9.0$ | $21.5 \pm 17.0$ | $54.7 \pm 14.6$ | $276.0 \pm 4.6$ | ✓ |
| EMMN | mushroom | $31.9 \pm 11.8$ | $13.5 \pm 12.4$ | $53.4 \pm 12.8$ | $275.4 \pm 4.6$ | ✓ |
| EMMN | rocket | $26.9 \pm 7.7$ | $24.3 \pm 23.7$ | $58.2 \pm 14.5$ | $274.8 \pm 3.9$ | ✓ |
| EMMN | sea | $30.3 \pm 8.8$ | $33.3 \pm 21.4$ | $69.1 \pm 16.0$ | $275.4 \pm 4.7$ | ✓ |
| BDSH | baby | $66.9 \pm 0.0$ | $16.9 \pm 0.3$ | $4.3 \pm 0.0$ | $85.2 \pm 1.1$ | ✓ |
| BDSH | lamp | $66.3 \pm 0.0$ | $16.7 \pm 0.5$ | $14.7 \pm 0.0$ | $85.4 \pm 2.2$ | ✓ |
| BDSH | mushroom | $66.8 \pm 0.0$ | $21.0 \pm 0.0$ | $12.6 \pm 0.0$ | $85.3 \pm 1.8$ | ✓ |
| BDSH | rocket | $66.2 \pm 0.0$ | $13.0 \pm 0.0$ | $2.9 \pm 0.0$ | $85.0 \pm 1.6$ | ✓ |
| Ours | baby | $67.1 \pm 0.0$ | $11.1 \pm 1.4$ | $9.9 \pm 0.3$ | $78.7 \pm 2.4$ | ✓ |
| Ours | lamp | $66.5 \pm 0.0$ | $32.4 \pm 0.7$ | $37.6 \pm 0.5$ | $79.4 \pm 2.8$ | ✓ |
| Ours | mushroom | $66.7 \pm 0.0$ | $41.4 \pm 1.1$ | $38.6 \pm 0.6$ | $79.6 \pm 3.1$ | ✓ |
| Ours | rocket | $66.2 \pm 0.0$ | $14.2 \pm 0.6$ | $2.9 \pm 0.3$ | $79.8 \pm 2.7$ | ✓ |
| Ours | sea | $66.6 \pm 0.0$ | $7.8 \pm 0.4$ | $7.9 \pm 0.3$ | $79.3 \pm 3.4$ | ✓ |

Table 10: VGG11 face unlearning class breakdown

| Method | Forget Class | $\mathcal{D}_r$ | $\mathcal{D}_f$ | MIA | Method Runtime | ZS |
|--------|------|------|------|------|------|------|
| Baseline | 1.0 | 94.0 ± 0.0 | 93.9 ± 0.0 | 13.8 ± 0.0 | 72.9 ± 0.5 | ✗ |
| Baseline | 10.0 | 94.0 ± 0.0 | 95.9 ± 0.0 | 11.5 ± 0.0 | 72.9 ± 0.8 | ✗ |
| Baseline | 20.0 | 94.1 ± 0.0 | 84.5 ± 0.0 | 65.2 ± 0.0 | 72.8 ± 1.0 | ✗ |
| Baseline | 30.0 | 93.9 ± 0.0 | 97.5 ± 0.0 | 11.2 ± 0.0 | 72.6 ± 0.8 | ✗ |
| Baseline | 40.0 | 94.0 ± 0.0 | 92.3 ± 0.0 | 15.4 ± 0.0 | 72.7 ± 0.6 | ✗ |
| Retrain | 1.0 | 100.0±0.0 | 0.0 ± 0.0 | 2.6 ± 0.8 | 491.9 ± 2.0 | ✗ |
| Retrain | 10.0 | 100.0±0.0 | 0.0 ± 0.0 | 0.7 ± 0.5 | 494.3 ± 1.8 | ✗ |
| Retrain | 20.0 | 100.0±0.0 | 0.0 ± 0.0 | 2.5 ± 1.0 | 492.7 ± 2.2 | ✗ |
| Retrain | 30.0 | 100.0±0.0 | 0.0 ± 0.0 | 6.0 ± 1.1 | 491.9 ± 2.7 | ✗ |
| Retrain | 40.0 | 100.0±0.0 | 0.0 ± 0.0 | 4.8 ± 1.5 | 494.7 ± 2.4 | ✗ |
| Finetune | 1.0 | 97.6 ± 0.7 | 36.9 ± 9.9 | 4.3 ± 2.7 | 84.7 ± 0.9 | ✗ |
| Finetune | 10.0 | 95.6 ± 2.9 | 43.8 ± 21.3 | 8.6 ± 6.7 | 84.5 ± 0.5 | ✗ |
| Finetune | 20.0 | 98.1 ± 0.5 | 25.1 ± 7.3 | 5.4 ± 1.8 | 84.8 ± 0.7 | ✗ |
| Finetune | 30.0 | 97.9 ± 0.8 | 23.8 ± 9.7 | 6.5 ± 3.7 | 84.8 ± 0.9 | ✗ |
| Finetune | 40.0 | 97.7 ± 0.6 | 16.6 ± 7.1 | 6.3 ± 3.2 | 84.5 ± 0.8 | ✗ |
| Amnesiac | 1.0 | 99.7 ± 0.1 | 0.0 ± 0.0 | 1.4 ± 1.3 | 82.2 ± 1.1 | ✗ |
| Amnesiac | 10.0 | 99.7 ± 0.1 | 0.0 ± 0.0 | 1.0 ± 0.8 | 82.2 ± 0.6 | ✗ |
| Amnesiac | 20.0 | 99.7 ± 0.1 | 0.0 ± 0.0 | 1.5 ± 0.6 | 82.1 ± 0.9 | ✗ |
| Amnesiac | 30.0 | 99.7 ± 0.1 | 0.0 ± 0.0 | 1.1 ± 0.7 | 82.1 ± 0.7 | ✗ |
| Amnesiac | 40.0 | 99.7 ± 0.2 | 0.0 ± 0.0 | 1.3 ± 1.2 | 81.9 ± 0.7 | ✗ |
| SCRUB | 1.0 | 98.8 ± 0.0 | 97.1 ± 0.0 | 8.8 ± 0.8 | 85.5 ± 0.8 | ✗ |
| SCRUB | 10.0 | 98.8 ± 0.0 | 97.5 ± 0.0 | 10.5 ± 0.3 | 85.5 ± 0.8 | ✗ |
| SCRUB | 20.0 | 98.8 ± 0.0 | 98.0 ± 0.3 | 74.2 ± 0.7 | 85.5 ± 1.1 | ✗ |
| SCRUB | 30.0 | 98.9 ± 0.0 | 95.9 ± 0.0 | 81.8 ± 0.9 | 85.6 ± 1.0 | ✗ |
| SCRUB | 40.0 | 98.9 ± 0.0 | 96.5 ± 0.5 | 78.0 ± 1.1 | 84.9 ± 0.8 | ✗ |
| SSD | 1.0 | 55.8 ± 0.0 | 0.0 ± 0.0 | 4.0 ± 0.0 | 92.0 ± 42.0 | ✗ |
| SSD | 10.0 | 73.7 ± 0.0 | 0.0 ± 0.0 | 2.5 ± 0.0 | 86.7 ± 27.2 | ✗ |
| SSD | 20.0 | 0.8 ± 0.0 | 0.0 ± 0.0 | 100.0 ± 0.0 | 78.2 ± 1.5 | ✗ |
| SSD | 30.0 | 86.0 ± 0.0 | 0.0 ± 0.0 | 10.1 ± 0.0 | 82.1 ± 10.0 | ✗ |
| SSD | 40.0 | 47.4 ± 0.0 | 0.0 ± 0.0 | 4.3 ± 0.0 | 78.2 ± 1.1 | ✗ |
| Teacher | 1.0 | 93.7 ± 0.3 | 0.0 ± 0.0 | 0.0 ± 0.0 | 75.0 ± 0.7 | ✗ |
| Teacher | 10.0 | 93.7 ± 0.3 | 0.0 ± 0.0 | 0.0 ± 0.0 | 74.6 ± 0.8 | ✗ |
| Teacher | 20.0 | 93.9 ± 0.3 | 0.0 ± 0.0 | 0.3 ± 0.4 | 74.7 ± 0.8 | ✗ |
| Teacher | 30.0 | 93.5 ± 0.3 | 0.1 ± 0.2 | 0.1 ± 0.2 | 74.5 ± 0.7 | ✗ |
| Teacher | 40.0 | 93.8 ± 0.2 | 0.4 ± 0.9 | 0.1 ± 0.3 | 75.0 ± 0.8 | ✗ |
| UNSIR | 1.0 | 99.5 ± 0.1 | 74.4 ± 9.2 | 13.6 ± 8.9 | 87.3 ± 0.5 | ✗ |
| UNSIR | 10.0 | 99.4 ± 0.3 | 87.1 ± 4.2 | 44.8 ± 9.2 | 87.7 ± 0.6 | ✗ |
| UNSIR | 20.0 | 99.4 ± 0.1 | 54.9 ± 15.2 | 10.3 ± 4.7 | 86.8 ± 0.9 | ✗ |
| UNSIR | 30.0 | 99.4 ± 0.1 | 65.7 ± 12.0 | 13.2 ± 5.2 | 87.8 ± 1.0 | ✗ |
| UNSIR | 40.0 | 99.5 ± 0.1 | 49.0 ± 23.9 | 6.9 ± 7.0 | 87.2 ± 0.8 | ✗ |
| GKT | 1.0 | 2.0 ± 0.6 | 0.0 ± 0.0 | 23.9 ± 30.2 | 441.5 ± 1.8 | ✓ |
| GKT | 10.0 | 2.0 ± 0.6 | 0.0 ± 0.0 | 53.4 ± 44.1 | 439.7 ± 3.1 | ✓ |
| GKT | 20.0 | 2.0 ± 0.6 | 0.0 ± 0.0 | 28.2 ± 34.8 | 441.4 ± 3.0 | ✓ |
| GKT | 30.0 | 2.1 ± 0.7 | 0.0 ± 0.0 | 28.4 ± 40.1 | 439.6 ± 3.4 | ✓ |

**Table 10 (Continued)**

| Method | Forget Class | $\mathcal{D}_r$ | $\mathcal{D}_f$ | MIA | Method Runtime | ZS |
|---|---|---|---|---|---|---|
| GKT | 40.0 | 2.1 ± 0.6 | 0.0 ± 0.0 | 44.9 ± 49.7 | 441.2 ± 2.3 | ✓ |
| EMMN | 1.0 | 51.0±13.5 | 69.3 ± 25.7 | 26.9 ± 17.8 | 283.6 ± 1.3 | ✓ |
| EMMN | 10.0 | 51.8±14.3 | 65.0 ± 21.8 | 27.8 ± 26.2 | 283.6 ± 1.9 | ✓ |
| EMMN | 20.0 | 41.4±12.9 | 26.2 ± 17.2 | 55.8 ± 13.5 | 282.9 ± 0.7 | ✓ |
| EMMN | 30.0 | 45.4±18.2 | 63.0 ± 27.7 | 32.9 ± 21.7 | 283.7 ± 1.2 | ✓ |
| EMMN | 40.0 | 46.6±17.0 | 26.5 ± 12.0 | 60.9 ± 7.4 | 282.8 ± 1.3 | ✓ |
| BDSH | 1.0 | 93.6 ± 0.0 | 79.4 ± 0.0 | 42.4 ± 0.0 | 91.24 ±1.7 | ✓ |
| BDSH | 10.0 | 94.0 ± 0.0 | 93.4 ± 0.0 | 79.5 ± 0.0 | 91.1 ± 0.7 | ✓ |
| BDSH | 20.0 | 93.7 ± 0.0 | 64.2 ± 0.0 | 32.9 ± 0.0 | 90.8 ± 1.8 | ✓ |
| BDSH | 30.0 | 94.1 ± 0.0 | 90.2 ± 0.0 | 65.9 ± 0.0 | 90.5 ± 3.5 | ✓ |
| BDSH | 40.0 | 94.0 ± 0.0 | 90.6 ± 0.0 | 73.5 ± 0.0 | 90.9 ± 1.5 | ✓ |
| Ours | 1.0 | 91.4 ± 0.1 | 1.9 ± 0.2 | 4.7 ± 0.5 | 74.1 ± 0.6 | ✓ |
| Ours | 10.0 | 93.0 ± 0.1 | 11.5 ± 1.8 | 4.8 ± 0.5 | 74.4 ± 0.9 | ✓ |
| Ours | 20.0 | 90.3 ± 0.1 | 8.2 ± 0.5 | 1.7 ± 0.0 | 74.2 ± 0.6 | ✓ |
| Ours | 30.0 | 91.6 ± 0.1 | 1.2 ± 0.0 | 5.0 ± 0.4 | 73.7 ± 0.7 | ✓ |
| Ours | 40.0 | 92.8 ± 0.1 | 35.1 ± 1.5 | 13.4 ± 0.6 | 74.3 ± 0.6 | ✓ |

Table 11: VGG11 CIFAR-20 sub-class unlearning breakdown

| Method | Forget Class | $\mathcal{D}_r$ | $\mathcal{D}_f$ | MIA | Method Runtime | ZS |
|---|---|---|---|---|---|---|
| Baseline | baby | 75.2 ± 0.0 | 82.3 ± 0.0 | 74.8 ± 0.0 | 76.0 ± 1.5 | ✗ |
| Baseline | lamp | 75.5 ± 0.0 | 56.9 ± 0.0 | 72.0 ± 0.0 | 75.3 ± 0.8 | ✗ |
| Baseline | mushroom | 75.3 ± 0.0 | 75.6 ± 0.0 | 73.4 ± 0.0 | 75.7 ± 0.8 | ✗ |
| Baseline | rocket | 75.3 ± 0.0 | 79.0 ± 0.0 | 83.0 ± 0.0 | 75.4 ± 1.1 | ✗ |
| Baseline | sea | 75.1 ± 0.0 | 92.9 ± 0.0 | 90.6 ± 0.0 | 75.0 ± 1.2 | ✗ |
| Retrain | baby | 72.6 ± 0.2 | 67.7 ± 3.0 | 55.4 ± 1.2 | 425.8 ± 1.4 | ✗ |
| Retrain | lamp | 73.0 ± 0.2 | 18.0 ± 2.8 | 17.9 ± 2.0 | 425.7 ± 1.6 | ✗ |
| Retrain | mushroom | 73.0 ± 0.3 | 8.8 ± 2.2 | 12.1 ± 1.3 | 425.6 ± 1.7 | ✗ |
| Retrain | rocket | 72.9 ± 0.2 | 11.5 ± 2.8 | 14.1 ± 1.3 | 425.5 ± 1.9 | ✗ |
| Retrain | sea | 72.6 ± 0.3 | 85.5 ± 2.6 | 66.8 ± 2.3 | 425.2 ± 1.7 | ✗ |
| Finetune | baby | 65.5 ± 0.8 | 60.5 ± 6.8 | 53.2 ± 5.9 | 120.3 ± 1.1 | ✗ |
| Finetune | lamp | 66.0 ± 1.6 | 17.5 ± 5.2 | 22.6 ± 4.1 | 121.2 ± 1.3 | ✗ |
| Finetune | mushroom | 65.7 ± 1.1 | 6.1 ± 6.0 | 18.5 ± 4.2 | 121.3 ± 0.6 | ✗ |
| Finetune | rocket | 65.5 ± 0.7 | 6.2 ± 3.7 | 22.3 ± 5.5 | 120.6 ± 0.9 | ✗ |
| Finetune | sea | 65.0 ± 0.8 | 82.2 ± 4.6 | 71.0 ± 9.6 | 120.4 ± 1.1 | ✗ |
| Amnesiac | baby | 73.7 ± 0.2 | 53.5 ± 6.4 | 5.0 ± 0.9 | 98.0 ± 1.1 | ✗ |
| Amnesiac | lamp | 74.2 ± 0.2 | 9.2 ± 2.4 | 7.1 ± 1.6 | 97.6 ± 0.9 | ✗ |
| Amnesiac | mushroom | 74.0 ± 0.2 | 2.5 ± 1.6 | 4.3 ± 0.9 | 97.8 ± 1.0 | ✗ |
| Amnesiac | rocket | 73.8 ± 0.2 | 2.4 ± 2.4 | 3.0 ± 0.9 | 98.4 ± 0.8 | ✗ |
| Amnesiac | sea | 73.7 ± 0.2 | 64.0 ± 17.7 | 1.2 ± 0.5 | 97.6 ± 1.2 | ✗ |
| SCRUB | baby | 75.0 ± 0.3 | 77.0 ± 2.3 | 71.0 ± 2.9 | 130.8 ± 2.8 | ✗ |
| SCRUB | lamp | 75.4 ± 0.2 | 41.2 ± 2.8 | 40.2 ± 6.7 | 130.4 ± 2.1 | ✗ |
| SCRUB | mushroom | 75.4 ± 0.1 | 15.5 ± 7.6 | 11.8 ± 0.8 | 130.8 ± 2.6 | ✗ |

**Table 11 (Continued)**

| Method | Forget Class | $\mathcal{D}_r$ | $\mathcal{D}_f$ | MIA | Method Runtime | ZS |
|---|---|---|---|---|---|---|
| SCRUB | rocket | 62.4±28.4 | 10.1 ± 22.5 | 16.7 ± 21.7 | 131.4 ± 3.8 | ✗ |
| SCRUB | sea | 75.0 ± 0.1 | 91.2 ± 1.6 | 83.1 ± 3.3 | 130.7 ± 2.5 | ✗ |
| SSD | baby | 72.0 ± 0.0 | 0.0 ± 0.0 | 8.8 ± 0.0 | 83.8 ± 1.0 | ✗ |
| SSD | lamp | 74.0 ± 0.0 | 3.6 ± 0.0 | 9.0 ± 0.0 | 84.4 ± 1.1 | ✗ |
| SSD | mushroom | 72.0 ± 0.0 | 0.0 ± 0.0 | 6.2 ± 0.0 | 84.0 ± 1.2 | ✗ |
| SSD | rocket | 75.0 ± 0.0 | 4.2 ± 0.0 | 11.0 ± 0.0 | 83.9 ± 0.8 | ✗ |
| SSD | sea | 74.8 ± 0.0 | 69.6 ± 0.0 | 56.2 ± 0.0 | 83.5 ± 1.4 | ✗ |
| Teacher | baby | 74.9 ± 0.2 | 77.9 ± 1.2 | 0.4 ± 0.7 | 78.7 ± 0.6 | ✗ |
| Teacher | lamp | 75.2 ± 0.1 | 35.5 ± 13.1 | 0.4 ± 0.4 | 79.5 ± 1.1 | ✗ |
| Teacher | mushroom | 75.0 ± 0.1 | 27.5 ± 14.5 | 0.2 ± 0.1 | 90.6 ± 36.9 | ✗ |
| Teacher | rocket | 74.9 ± 0.2 | 48.4 ± 16.9 | 0.1 ± 0.1 | 91.5 ± 38.0 | ✗ |
| Teacher | sea | 74.7 ± 0.1 | 85.3 ± 2.6 | 0.1 ± 0.1 | 78.4 ± 0.9 | ✗ |
| UNSIR | baby | 73.8 ± 0.2 | 77.7 ± 2.1 | 77.6 ± 2.5 | 110.8 ± 0.9 | ✗ |
| UNSIR | lamp | 74.2 ± 0.3 | 44.5 ± 7.4 | 55.1 ± 6.2 | 110.7 ± 0.7 | ✗ |
| UNSIR | mushroom | 74.0 ± 0.3 | 47.1 ± 4.0 | 41.7 ± 8.2 | 110.6 ± 1.3 | ✗ |
| UNSIR | rocket | 74.1 ± 0.2 | 57.5 ± 10.2 | 57.4 ± 8.6 | 111.3 ± 0.7 | ✗ |
| UNSIR | sea | 73.8 ± 0.2 | 90.8 ± 1.8 | 85.3 ± 3.8 | 109.5 ± 1.0 | ✗ |
| BDSH | baby | 74.2 ± 0.0 | 59.1 ± 0.0 | 31.2 ± 0.0 | 86.0 ± 1.1 | ✓ |
| BDSH | lamp | 75.3 ± 0.0 | 36.5 ± 0.0 | 36.8 ± 0.0 | 85.2 ± 1.3 | ✓ |
| BDSH | mushroom | 75.06±0.0 | 36.6 ± 0.0 | 25.7 ± 0.0 | 86.5 ± 2.5 | ✓ |
| BDSH | rocket | 74.4 ± 0.0 | 17.5 ± 0.0 | 12.9 ± 0.0 | 85.9 ± 1.6 | ✓ |
| BDSH | sea | 72.9 ± 0.0 | 39.1 ± 0.0 | 11.8 ± 0.0 | 85.3 ± 1.2 | ✓ |
| Ours | baby | 73.6 ± 1.3 | 49.8 ± 19.4 | 33.3 ± 17.5 | 79.5 ± 3.6 | ✓ |
| Ours | lamp | 74.0 ± 1.3 | 32.5 ± 15.8 | 32.4 ± 19.8 | 79.7 ± 3.2 | ✓ |
| Ours | mushroom | 74.2 ± 0.7 | 33.7 ± 12.3 | 23.5 ± 12.8 | 79.1 ± 4.0 | ✓ |
| Ours | rocket | 73.7 ± 0.8 | 19.3 ± 18.3 | 11.2 ± 7.8 | 80.1 ± 3.3 | ✓ |
| Ours | sea | 73.1 ± 1.0 | 37.8 ± 22.1 | 17.3 ± 20.2 | 79.0 ± 3.2 | ✓ |

