# OpenReview forum: "An Information Theoretic Approach to Machine Unlearning"
_TMLR — Accepted by TMLR_

### Review · Reviewer_uvPP · 2025-01-10

**Summary Of Contributions:**

The authors introduce JiT, which is a zero-shot approach to unlearning that does not require access to the retain dataset (can be impractical in some scenarios). The method main advantage is performance as it can obtain (close to) state-of-the-art performance while requiring less compute power than previous techniques. The method is evaluated in image classification tasks.

**Audience:**

Yes

**Claims And Evidence:**

Yes

**Requested Changes:**

Major requested changes:
* See W1. Clarify the zero-shot specification in this case and why access to forget dataset is granted, unlike the cited work.
* See W2. Include a detailed analysis of the effectiveness on samples that leak the most information. I could see this as: (1) look for samples that leak the most information (e.g. most confident in MIA attack), (2) try to unlearn them with all methods, (3) report performance after unlearning.
* The authors claim

Minor changes:
* It would be great to detail how to interpret the MIA score.
* Do the authors have any insights on how the method works with increasing input dimensionality? My intuition is that higher dimensionalities may make decision boundaries harder to characterize. This is relevant since the scale at which storing the retain dataset becomes challenging may use very high dimensional data.

Other suggestions:
* Tables are now a bit hard to read. This might not be necessarily a good idea but I wanted to share with authors so that they can consider if useful somewhere. The evaluation is mostly about seeing "how close methods are to a retrained model". Then, I would suggest that authors can report the metric on baseline and retrain and then report the delta of the other methods with respect to retrain. Additionally, with our without doing the delta change, you could color in red or green depending on whether the difference with respect to baseline is "good" or "bad". Namely, if accuracy on retain is higher than baseline then green but if accuracy on forget higher then red.

**Strengths And Weaknesses:**

# Strengths

* The method is well motivated in terms of the information gain that each data point has.
* The formulation in terms of data points offers a more generic method that can target single points rather than entire classes.
* The evaluation is comprehensive in terms of models, methods and datasets; covering most metrics and comparisons with SOTA methods.
* JiT obtains comparable performance to methods that require access to the retain dataset in the average case. This can be relevant if storing retain data is challenging.


# Weaknesses

* W1: The original work by Chundawat et al. (2023) [1] introduces Zero-shot MU as a **data-free** method that does not require access to either retain or forget datasets (see Section IV, A). The authors of this paper define ZS unlearning as having access to both the model and the forget dataset.
* W2: When it comes to privacy, one might be interested in removing worst-case effects (i.e. samples that leak the most information). These samples are often outliers. All evaluations in the paper focus on average in-distribution unlearning.

[1] Chundawat, Vikram S., et al. "Zero-shot machine unlearning." IEEE Transactions on Information Forensics and Security 18 (2023): 2345-2354.

# Overall assessment

I think this paper will be of interest to the unlearning community. One of the main limitations is the lack of analysis on worst-case samples, but I believe authors could include this easily during the revisions.

---

> ### Author Response · Authors · 2025-01-28
> **Response to Reviewer uvPP Comments**
>
> Thank you for these important suggestions for improvement. Please find our point-by-point response below:
>
> 1. Regarding the zero-shot specification and access to forget data: We acknowledge that our definition of 'zero-shot' requires clarification, particularly in relation to prior work. The zero-shot formulation in Chundawat et al precludes the use of arbitrary forget sets, since no access to forget data is allowed. This limits their zero-shot setting to operate at a class level and require class-type mappings, in our setting we generalize this to handle realistic scenarios that require unlearning arbitrary forget sets, including individual samples, but this requires access to the forget set. We consider this “zero-shot” as we assume no access to any other data, and no access to any previous model states. We have updated the manuscript to highlight this distinction.
>
> 2. Regarding the analysis of information leakage: We appreciate this excellent suggestion for a more thorough evaluation. We have added the recommended experiment. Specifically, we identified high-information-leakage samples using membership inference attacks (MIA) on the 'rocket' class in our dataset. We collected the 50 samples with highest MIA confidence scores (indicating strongest information leakage) and applied unlearning methods to these worst-case samples. The results demonstrate our method's effectiveness even in these challenging cases where information leakage is most prominent. We have added a detailed section presenting these results in the revised manuscript.
>
> 3. The MIA score corresponds to the percentage of forget-set samples that a membership inference attack successfully identified as belonging to the original training set. In our case, the MIA is a logistic regression trained over the entropy of the classifier on a subset of the original train/test data.
>
> Thank you again for your review

---

### Review · Reviewer_vGoy · 2025-01-16

**Summary Of Contributions:**

The article presents a novel approach to data unlearning for neural networks in a zero-shot context, where the original training data is unavailable. This method relies on the informativeness of the data being removed—specifically, whether the information carried by the removed samples can be inferred from the remaining dataset / neighboring data points in the feature space. The paper starts with outlining the method's underlying intuition and interpretation before evaluating its empirical performance against various benchmarks using image datasets like CIFAR10. Additionally, it provides a detailed analysis of the results and discusses the method's limitations.

**Audience:**

Yes

**Claims And Evidence:**

Yes

**Requested Changes:**

See weaknesses section.

**Strengths And Weaknesses:**

Strengths:
- Paper is well structured and very clear
- Extensive discussion of the intuition behind the method, its performances and limitation
- Approach based on informativeness of data is interesting and empirical results presented look good
- Information-based approach is intuitive and generalizable to multiple types of problems


Weaknesses:
- Paper could gain at being more formal, especially when defining the method - and section 4 in general
- Fairly limited empirical study on image data only - I would have liked to see the approach tried on different types of data
- More generally, analyzes of the paper can be deepened

---

> ### Author Response · Authors · 2025-01-28
> **Response to Reviewer vGoy Comments**
>
> Thank you for your thoughtful review and positive comments on our paper's structure, intuition, and information-based approach. We appreciate the constructive feedback on areas for improvement. While your summary indicates overall satisfaction with the paper, we would welcome specific suggestions about:
> 1. Which parts of Section 4 would benefit from increased mathematical formality.
> 2. Which analyses you feel could be deepened.
>
> In addition, we invite you to review the improvements to formality and analysis that we have made in response to comments from other reviews (please see our other rebuttal comments). If you feel there are areas of further improvement, we are happy to make the additional changes.
>
> Thank you again for your review.

---

### Review · Reviewer_rKHL · 2025-01-16

**Summary Of Contributions:**

The paper presents a novel approach to machine unlearning that is inspired by information theory. The authors show empirically that the method is the fastest Zero-Shot method, and it even competes with non-zero-shot techniques in some settings. They also provide a discussion of the geometric intuitions behind their method. Finally, they show in a range of experiments that their method outperforms several baselines.

**Audience:**

Yes

**Broader Impact Concerns:**

None.

**Claims And Evidence:**

No

**Requested Changes:**

## Critical
1. The math terminology and definitions need to be revised.
    1. Smoothness, curvature, gradient norm, etc. should all be used appropriately and with care to specify what functions are being differentiated and with respect to which variables.
    2. The claims about what the loss looks like near boundaries needs to be clarified, corrected, or supported.
2. Figures 3 and 4 need axes labels and units.
3. Experiments should be extended to include more modern architectures and datasets. (While better choices exist, even seeing ResNets and using ImageNet data would be good.)

## Strengthen the Work
1. A bit more related work on unlearning would help contextualize the work. I realize this isn't about generative modelling, but much of the unlearning work in the last year or so is on LLMs and some mention of that work and why this is still relevant or how it is different from those methods and evaluations would be good.
    1. RMU (Li, Nathaniel, et al. "The wmdp benchmark: Measuring and reducing malicious use with unlearning." arXiv preprint arXiv:2403.03218 (2024).)
    1. NPO  (Zhang, Ruiqi, et al. "Negative preference optimization: From catastrophic collapse to effective unlearning." arXiv preprint arXiv:2404.05868 (2024).)
    1. TOFU (Maini, Pratyush, et al. "Tofu: A task of fictitious unlearning for llms." arXiv preprint arXiv:2401.06121 (2024).)
2. A mention of randomized smoothing is missing. Several works on adversarial robustness use the same principle that this method is based to make models robust [Cohen, Jeremy, Elan Rosenfeld, and Zico Kolter. "Certified adversarial robustness via randomized smoothing." international conference on machine learning. PMLR, 2019.]. This should be mentioned and would better motivate the work as a localized version of randomized smoothing for unlearning.

## Minor issues that don't affect my recommendation but would help the paper
1. Typo in the second paragraph. "As such, Chundawat et al. (2023b introduces..." should be "introduce" as "...et al." is a plural group.
2. On Page 2, there is a sentence that doesn't make sense: "This minimal change can be effectively measured via the curvature of the model should be low in the region surrounding such a point."
3. In the first line on Page 3, the citation of Lindley (1956) should be in parentheses.
4. In Table 1, the acronyms for baseline, retrain, and finetune are making it harder for the reader. These somewhat non-standard shorthand names require double checking the text and save little to no space in the table. My recommendation is that the words "Baseline" and "Retrain" etc. appear directly in the table to save the reader from having to translate/look up these codes.

**Strengths And Weaknesses:**

# Strengths:
1. The motivation that some data contain more information than others and that unlearning methods should respect this is sound and compelling.
2. The method performs well on the benchmark tasks.

# Weaknesses:
1. Some confusing (possibly misused) vocabulary
    1. "Curvature" might be misused in this paper. If it is the correct choice, then the language around it is confusing. What I mean is that from the math and the description of the method, it seems the focus is on the gradient at the forget points and in the surrounding neighborhood. Equation (1), for example, is about minimizing the local gradient. Generally, curvature is a second order phenomenon that describes how flat or not flat a surface or a curve is, and gradients are a first order phenomenon that discuss the local rate of change. These are fundamentally different values as a steep line or plane could have large gradient norms (or a steep slope) but still have zero curvature. The word "curvature" is used several times in the Section 1 (Page 2), but never discussed in the technical sections.  Furthermore, in Section 1, there is no description of which space is being discussed. At this point in the paper, it is unclear if the authors are discussing curvature or slope of the loss with respect to parameters (thus discussing parameter space) or with respect to inputs (input space).
    1. "Entropy of the data" is discussed but no formula is given. To discuss entropy requires a probability distribution and I can't find that in the paper. Figure 4, for example, reports $\mathcal{H}(x)$ but I don't see how that is computed. Is there a function or approximation of $P(X|Y)$ that is used to measure the entropy of the data?
        1. "Intuitively, low-entropy predictions indicate higher model confidence, and therefore we expect that the entropy of the model after JiT unlearning is applied will be higher, aligning closely with that of the retrained model." This is the first sentence that indicates to me that "entropy of the data" refers to the entropy of the logits at a point. If this is what is always meant, it should be much more clear earlier (listed below in Requested Changes).

2. Incorrect of unsupported mathematical statements.
    1. Definition 4.2 defines $\mathcal{B}_r(x)$ as a neighborhood but then says "... let $Y$ be a random variable corresponding to a sample $\mathcal{B}_r(x)$ belonging to class $C$." Is $\mathcal{B}_r(x)$ a neighborhood (i.e. a bounded region of space) or a sample?
    1. The language in the paragraph before Equation (1) makes clear that by "low information" the authors mean data whose class can be inferred from nearby points. The definition of "low information" in Definition 4.2, however, says it has to do with the conditional entropy of the data given its class. I don't think the class label and the local neighborhood are correctly used in these definitions. It seems to me that a definition of low information that would be consistent with the rest of the paper has to do with the distribution of class labels among nearby points. If the neighborhood around a sample is dominated by similarly labeled data, we might conclude that it has little bearing on the final decision boundary. So the information content of $x$ with respect to the classifier is related to $P(Y|x' \in \mathcal{B}_r(x))$ or something similar.
    1. "...if the classifier is smooth with respect to a forget sample.." (in the paragraph before Equation (1)) is misusing the terminology. A function can be smooth "at a point" "in a neighborhood" or "with respect to an input variable". I assume the authors mean here that the function is locally smooth with respect to the inputs in the neighborhood of the particular forget sample. This phrasing should be corrected (listed below in Requested Changes). More important to this section of the review is that "smoothness" isn't the quantity measured or discussed elsewhere. Given that the classifier is parameterized by a neural net, it is differentiable. It's smoothness (as determined by the continuity of its higher order derivatives) isn't discussed elsewhere. Is this what the authors meant to mention here?
    1. "...then the model's prediction over this sample can be viewed as being interpolated from other data." This claim is unsupported. Whether the authors are discussing gradient norm, curvature, or smoothness (all distinct mathematical properties), the value of these measurements at a point in input space of a trained neural network may or may not result from nearby training data.
    1. "minimising the gradient of the classifier with respect to the forget set" This is another misuse of the phrase "with respect to." Equation (1) minimises the gradient w.r.t the inputs *around* points in the forget set.
    1. "By definition, the model will experience a large rate of change at the decision boundary," (in the last paragraph of Section 4.1). This is not true. There are two functions that could be discussed here, the discontinuous integer-valued classifier that maps inputs to a discrete class label (implementationally, this is the function defined by taking the argmax of the logits) *or* the continuous, real-valued function that maps input points to scalar loss values. If the authors are talking about the discontinuous classifier, then there is a discontinuity at the decision boundary by definition--not a "large rate of change." If the function being discussed here is the continuous  and differentiable loss function (which is consistent with the rest of the paper), then there is no support for the claim that it has a high rate of change at the decision boundary. The decision boundary is where the argmax of the logits changes, but the loss may vary very slowly there. In more technical terms, the gradient norm is not bounded in the neighborhood of a decision boundary. Furthermore in the math that follows this sentence, I think it's incorrect to call the unit vectors "noise" if they are specifically in the direction of the boundary. These are not noise or random in any sense, they are particular unit vectors determined fully by the function and the point $x$.
3. Confusing Figures
    1. Figure 2 (Very Minor): the color choice made it hard for me to find the forget sample. Also, curvature is mentioned in the caption with no indication of the curvature in the plots. (Also, see above, I suspect this isn't really about curvature).
    1. Figure 3: What is the experiment here? What is the parameterization of the learned model? Are the data points actually from a sigmoid function? Which regions of space are you referring to as "high curvature"? The largest changes from red to black occur at -2  and 2 on the horizontal axis (unlabeled). This is not where the derivative of the red curve is highest, is it supposed to be? The outliers here at -4 and 4 on the horizontal axis seem not to change much, but the intuition from prior sections is that proximity to similarly labeled data is what should matter. Furthermore, the rest of the work is about classifiers, but here is about regression and this is confusing with no context.
    1. Figure 4: The vertical axis isn't labeled with a variable or units.
4. Experiments on outdated architectures: why are the authors considering VGG (published about 10 years ago)? There are much more relevant modern/recent architectures.

---

> ### Author Response · Authors · 2025-01-28
> **Response to Reviewer rKHL Comments part 1**
>
> We thank the reviewer for their detailed and valuable comments on our work. We would like to offer the following comments and updates regarding the suggestions outlined in the review.
>
>
> 1.  After careful consideration we agree that the word curvature should not be used throughout our work. Instead we have updated the paper to discuss the rate of change of the model with respect to the input, or the gradient. To clarify, regarding the vector spaces being discussed in section 1, we care about the rate of change of the softmax classification with respect to variation in the input space.
>
> 2. Thank you for this important clarification request – Let us clarify. For a given input x \in X, our model f_\theta(x) produces a probability distribution over the set of classes C = {1, …, k}, where each f(x)_i represents the probability of class i. We then compute the entropy H of the models predictions for a given input x as:
>
>     $H(x) = - \sum_{i = 1}^k f(x)i \log(f(x)i)$
>
>     When we refer to ‘entropy of the data’, we are specifically measuring the entropy of the model's predicted probability distribution over classes for each input data point. The higher entropy after JiT unlearning indicates increased uncertainty in the model's predictions, as the model becomes less confident in its classifications - similar to what we observe in a fully retrained model that has properly forgotten the requested data.
>
> 3. Thank you for identifying this inconsistency. We confirm that Bᵣ(x) specifically denotes a neighborhood in the input space - a bounded region around point x with radius r. Definition 4.2 contained a typo in its second reference to Bᵣ(x).
>
> 4. We have updated the definition 4.2 to a more concise definition for information that is consistent with the rest of the paper and that corresponds to the distribution of class labels among nearby points.
>
> 5. We agree that our use of 'smooth' was imprecise and potentially misleading. We have revised the text to more accurately reflect what we are measuring: the local gradient magnitude of the classifier's output with respect to input perturbations. Instead, we will refer to regions being ‘'locally flat”, meaning that ||∇ₓf(x)|| is small in the neighborhood of the forget sample x, where f represents our classifier's output. This characterizes a region where small perturbations to the input produce minimal changes in the model's predictions. This directly connects to our objective function in Equation (1), which explicitly minimizes these gradients. We have updated the paragraph before Equation (1) to read: '...if the classifier exhibits small output variation with respect to input perturbations around a forget sample (i.e., has small local gradients), then this sample likely has minimal influence on the decision boundary...'
>
> 6. We thank the reviewer for raising this issue. We would like to extend our assumptions about the neural network being analysed. Specifically, we assume that as the number of samples within the neighborhood considered approaches infinity then the point interpolates. We thank the reviewer for identifying this unsupported claim. You are correct - our statement about interpolation requires additional assumptions to be mathematically sound. We propose the following refinement:
>     Under the following conditions:
>     * Let f be our neural network classifier
>     * Consider a point x₀ and its ε-neighborhood B_ε(x₀)
>     * Assume ||∇f(x)|| ≈ 0 for all x ∈ B_ε(x₀)
>
>     Then, as the sampling density n → ∞ in B_ε(x₀), f(x₀) can be approximated by a local interpolation of neighboring points, as the dense sampling and near-zero gradients imply local constancy of the function.
>
> 7. Thank you for this correction. You are right - our phrasing was imprecise. We will revise the text to accurately reflect what Equation (1) minimizes: 'minimizing the gradient of the classifier with respect to inputs generated by perturbing points in the forget set.'
>
> 8. You are correct that our statement about 'large rate of change at the decision boundary' was mathematically imprecise. While high gradient norms are not theoretically guaranteed at decision boundaries, empirical studies have observed this behavior in well-trained neural networks. As shown in [1], neural networks tend to learn functions that exhibit higher frequency components between classes and lower frequency components within classes. We have experimentally found that high-accuracy models learn a function that is high frequency in regions between different image classes but low-frequency within each class, validating the intuition that a “good” model should have sharp decision boundaries to delineate different classes, but smooth behavior within each class. However, we acknowledge this is an empirical observation rather than a mathematical necessity, and hence we have weakened our claim.
>
> [1] Spectral Bias in Practice: the Role of Function Frequency in Generalization, Fridovich-Keil et al. Neurips 2022

---

> ### Author Response · Authors · 2025-01-28
> **Response to Reviewer rKHL Comments part 2**
>
> 9. Thank you for the detailed observations about our figures. We have implemented the following improvements:
>     Figure 2: Adjust color scheme to highlight forgotten sample and remove incorrect 'curvature' terminology from caption.
>
>     Figure 3: The experiment shows a binary classification problem in 1D, where we use a simple neural network with a single weight and bias, followed by a sigmoid activation: S(x) = σ(wx + b). The red curve shows the initial decision boundary learned on the training data. The black points represent the model's output after applying our unlearning method to specific points. This simplified 1D example illustrates a key intuition of our method: points far from the decision boundary (e.g., at x = -4 and x = 4) naturally have low gradient magnitudes ||∇ₓS(x)|| in their neighborhoods, as the sigmoid function approaches its asymptotes. These regions correspond to high-confidence predictions, where the model is most certain about the class assignment; this is exactly the intuition behind our method. Finally, we have updated this plot to include axis labels.
>
>     Figure 4: To improve clarity, we have replaced the ridge plot with a boxplot that has clear units and axis labels.
>
> 10. We appreciate the reviewer's suggestion about modern architectures and datasets. We would like to highlight that our work already includes experiments with contemporary architectures and large-scale datasets. We show performance on a state-of-the-art architecture (ViT) for all experimental setups and, as shown in Table 5, we demonstrate our method's effectiveness on a 300m parameter ViT-Large using the ImageNet dataset, specifically comparing our approach against other zero-shot methods on this large-scale, modern benchmark. Additionally, we have added experiments with the ResNet-18 architecture, showing good performance forgetting a random subset of data.
>
> 11. We thank the reviewer for identifying additional related work. We have expanded our related work section with a discussion around unlearning generative models as well as a brief section linking our method to randomized smoothing from the adversarial robustness literature.
>
> 12. Thank you for identifying these minor clerical issues, we have worked to correct them where appropriate.
>
> Thank you again for your detailed review and valuable comments.

---

### Author Response · Authors · 2025-01-28
**Summary of Changes**

We thank the Action Editor and Reviewers for their time, and their detailed reviews. Below is a summary of the changes made to the manuscript. A revised version has now been uploaded to OpenReview.

1. We have made extensive changes to the notation and mathematical language to ensure an accurate and consistent description of our method (please see our comments to reviewer rKHL for specifics)
2. We have included additional sections in related work to discuss unlearning for generative models, as well as our connection to randomized smoothing from adversarial robustness.
3. We have updated several figures to ensure they have the correct labels and units, as well as ensuring clarity and ease of reading.
4. We have added an additional experiment with ResNet18.
5. We have clarified our zero-shot setting in relation to that of Chundawat et al, highlighting the distinctions and why they are necessary to generalize the setting to realistic scenarios.
6. We have conducted an additional experiment that explores how each unlearning algorithm performs when forgetting the highest information-leakage samples (i.e. the "worst-case scenario"). Please see discussion with reviewer uvPP for details.
7. We have made a range of minor clerical corrections.

---

### Decision · Action_Editor_qjgm · 2025-02-25

**Recommendation:** Accept with minor revision

**Comment:**

The idea in this paper is novel and should be of interest to some of the TMLR audience. The authors should complete the writing improvements suggested by the reviewers (improve the figures, improve the math terminology, improve table readable). The authors should also expend their Related Works section to include the suggestions from reviewer rKHL. The authors should also address to the two major requested changes from reviewer uvPP.

**Audience:**

This paper will be of interest to those studying unlearning in neural network systems.

**Claims And Evidence:**

The paper proposes a novel zero-shot unlearning method, and they demonstrate its efficacy on a variety of image classification tasks, including facial recognition (PinsFaceRecognition) and CIFAR-20. Two of the reviewers expressed interest in seeing additional evidence of the method's efficacy on a greater number of architectures and datasets.